# Balancing Fairness and Accuracy in Data-Restricted Binary Classification

## Abstract

Applications that deal with sensitive information may have restrictions placed on the data available to a machine learning (ML) model. For example, in some applications a model may not have direct access to sensitive attributes. This can affect the ability of an ML model to produce accurate and fair decisions. This paper proposes a framework that models the tradeoff between accuracy and fairness under four practical scenarios that dictate the type of data available for analysis. In contrast to prior work that examines the outputs of a scoring function, our framework directly analyzes the joint distribution of the feature vector, class label, and sensitive attribute by constructing a discrete approximation from a dataset. Through formulating multiple convex optimization problems, we answer the question: *How is the accuracy of a Bayesian oracle affected in each situation when constrained to be fair?* Analysis is performed on a suite of fairness definitions that include group and individual fairness. Experiments on three datasets demonstrate the utility of the proposed framework as a tool for quantifying the tradeoffs among different fairness notions and their distributional dependencies.

## 1 Introduction

A variety of studies have found bias to exist in machine learning (ML) models (Sweeney (2013); Angwin et al. (2016); Larson et al. (2016); Buolamwini & Gebru (2018); Larson et al. (2017)), raising concerns over their use in high-stakes applications. For example, Angwin et al. (2016) and Larson et al. (2016) found that a tool used to calculate the risk of criminal defendants repeating a crime was biased against African Americans. To address these concerns, a variety of mathematical definitions have been constructed to quantify the fairness of such models, the most prominent of which fall under two categories—group fairness (Kamiran & Calders (2012); Hardt et al. (2016); Chouldechova (2017); Zafar et al. (2017); Pleiss et al. (2017)) and individual fairness(Dwork et al. (2012); Petersen et al. (2021)). Group fairness definitions aim to quantify biases that may exist among the results produced for two or more demographic groups, while individual fairness focuses on ensuring the fair treatment of similar individuals. However, multiple theorems have shown the impossibility of satisfying multiple fairness definitions simultaneously. This motivates the following questions—To what extent can a model simultaneously satisfy multiple definitions of fairness exactly or in some relaxed form? If a model is capable of simultaneously satisfying multiple fairness definitions, what cost must it pay in terms of accuracy?

We should also be aware that the answers to such questions depend on the information to which a model has access. In the simplest case, a model is allowed to incorporate sensitive attributes in the decision-making process. However, it is often the case in dealing with applications that involve sensitive information that limitations are placed on the data available to a model. For example, financial institutions are not allowed to ask applicants their race when applying for loans, but must prove that their decisions are anti-discriminatory with respect to any sensitive attribute listed by the Fair Housing Act and Equal Credit Opportunity Act (Congress (1968; 1974-10)). Similarly, the Civil Rights Act of 1964 (Berg (1964)) requires that higher education institutions do not discriminate against applicants on the basis of a variety of sensitive attributes, including race, sex, and religion. However, analyzing the equity of their decisions is not always directly possible since it is not mandatory for applicants to provide such demographic information in their applications. Such situations are captured by the definition of unawareness proposed by Kusner et al. (2017). In certain situations, a model is only permitted to use features that have been decorrelated with

respect to the sensitive attribute to make decisions. Financial institutions, for example, commonly form separate ML and compliance teams in the same institution. Compliance teams oversee the handling of sensitive information and are responsible for ensuring that the decisions produced by a company are non-discriminatory, and ML teams train models to produce decisions for a company. While a compliance team is provided with all sensitive attribute information, when available, an ML team is prohibited from accessing such sensitive information and should not be able to deduce it from the data (de Castro et al. (2020)). In other words, the features provided to the ML team must be decorrelated with respect to the sensitive attribute, a notion introduced by Zemel et al. (2013).

Motivated by these circumstances, the focus of this paper is to analyze the tradeoff in accuracy that a baseline model incurs when it is required to satisfy multiple fairness notions under different situations that limit the data available to a model. Fig. 1 provides an overview that characterizes the main modules of our analysis. We aim to directly analyze the joint distribution of the feature vector, sensitive attribute, and class label, $(X, A, Y)$, as seen in Fig. 1a. In reality, we do not have access to this distribution, but rather a sampling of it in the form of a dataset. A discrete approximation of this joint distribution can be constructed by applying vector quantization (VQ) (Gersho & Gray (1991)) to a dataset and accumulating the statistics within each VQ cell. Since the number of samples in a VQ cell may be small, we densely sample a generator that has learned to latent structure of this joint population distribution to faithfully construct a fine-grained discrete approximation to it, $(\tilde{X}, \tilde{A}, \tilde{Y})$.

In our subsequent analysis, we derive an optimization formulation to model the behavior of an idealized classifier, referred to as the Bayesian oracle, when it is constrained to satisfy a various fairness definitions. The Bayesian oracle is designed to be stochastic, and is thus modeled as a scoring function, $S$, that assigns scores to different feature vectors, reflecting their probability of receiving a positive class label. All analyzed fairness definitions can be directly encoded as constraints in our framework, allowing us to avoid proactively constructing objective functions that indirectly satisfy them. While commonly analyzed in isolation, we concurrently formulate constraints from individual and group fairness definitions to investigate their relationship. Our analysis is performed under four data-restricting situations listed in Fig. 1b., under which the sensitive attribute **is (is not)** available and the features used for classification **are (are not)** required to be decorrelated from it. We elaborate on these situations in Section 3. Experiments conducted on three public datasets reveal that our framework captures the distributional dependence of the tensions that exist between different fairness notions and suggest that coupling individual and group fairness prevents the Bayesian oracle from arbitrarily penalizing individuals to satisfy group fairness. We also observe that a fair Bayesian oracle is typically able to maintain its accuracy on feature vectors that have been decorrelated from the sensitive attribute.

The remainder of this paper is organized as follows. Section 2 provides the definitions of fairness that we analyze in our tradeoff analysis along with the restrictions place on the data for each analysis scenario. In Section 3, we formulate a framework for analyzing the reduction in accuracy incurred from forcing a Bayesian oracle to be fair under four scenarios that contrain the information available to a Bayesian oracle. Experimental results are presented in Section 4 to quantify the tradeoffs between accuracy and fairness under each of these scenarios. Finally, we conclude the paper and provide discussion in Section 5.

**Related Work:** Multiple works have shown the impossibility of satisfying multiple definitions of fairness (Chouldechova (2017); Kleinberg et al. (2016); Zhao & Gordon (2022)). For example, Chouldechova (2017) showed that attempting to exactly satisfy three group fairness definitions simultaneously is futile, while Kleinberg et al. (2016) showed that demographic parity and equalized odds are in conflict when demographic groups have unequal class label balance. This has motivated a number of studies to analyze the tradeoffs between the accuracy and fairness of ML outcomes, the majority of which analyze a single fairness notion using pre-processing (Calmon et al. (2017); Kamiran & Calders (2012); Luong et al. (2011)), in-processing (Chen & Wu (2020); Jiang et al. (2020); Zafar et al. (2017)), or post-processing (Menon & Williamson (2018); Petersen et al. (2021); Lohia et al. (2019)) methods. Other studies have been conducted to analyze the tradeoff between accuracy and multiple fairness

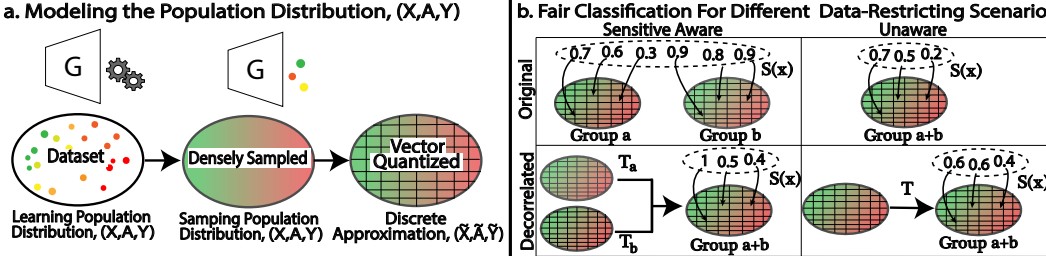

Figure 1: Outline of proposed framework for tradeoff analysis. (a) A discrete approximation of the population distribution is constructed by using a generator to densely sample it and applying vector quantization. (b) The accuracy-fairness tradeoff is analyzed under four data-restricting situations in which the sensitive attribute **is (is not)** available and the features used for classification **are (are not)** required to be decorrelated from it.

definitions (Kim et al. (2020); Liu & Vicente (2022); Hsu et al. (2022); Celis et al. (2019)). Kim et al. (2020) analyzed the tradeoffs between accuracy and fairness by constructing primarily linear constraints from the joint distribution between the class label and sensitive attribute of a dataset, though they do not consider how correlations with the features affects this tradeoff. Liu & Vicente (2022) propose an in-processing method for analyzing the tradeoff between accuracy and multiple fairness definitions, though they are required to use proxy fairness constraints to avoid solving a non-convex optimization problem. Hsu et al. (2022) analyze the tradeoff between accuracy and multiple fairness definitions through post-processing the scores produced by a model. However, such an approach may provide less flexible analysis since the transformation from the space of feature vectors to the space of scores is many-to-one, which could force many non-similar feature vectors to be treated similarly. Overall, none of these works incorporate both individual and group fairness notions into their analysis, nor do they directly analyze how the limitations on the information available to a model may affect their analyses.

## 2 Preliminaries for Tradeoff Analysis

The following setup is used to explicitly formulate each definition analyzed in our framework. Let $X, A$, and $Y$ represent the random feature vector and sensitive attribute and class label random variables respectively associated with the sample spaces $\mathcal{X}, \mathcal{A}$, and $\mathcal{Y}$. For simplicity, assume that $\mathcal{X} = \mathbb{R}^k$, $\mathcal{A} = \{a, b\}$, and $\mathcal{Y} = \{0, 1\}$, though our formulations may be generalized to non-binary sensitive attributes through combinatorial extension. Let $S : \mathcal{X} \to [0, 1]$ be a randomized scoring function whose output represents the conditional probability with which we assign a feature vector a label of 1, which we refer to as a score. Its associated randomized estimator is:

$$\hat{Y}(x) = \begin{cases} 1 & , \text{ w.p. } S(x) \\ 0 & , \text{ w.p. } 1 - S(x) \end{cases}.$$

**Fairness Defintions**  Different fairness definitions proposed in the literature provide tools for ensuring that ML models uphold various societal values. In our framework, we analyze the following suite of fairness definitions: Demographic Parity (DP), Equal Accuracy (EA), Equal Opportunity (EOp), Predictive Equality (PE), Equalized Odds (EOd), and Local Individual Fairness (Ind) (see Appendix B for a list of their explicit definitions).

**Data-Restricting Definitions**  A model's ability to balance accuracy and fairness depends on the data available to it. We provide two practical definitions that dictate the data available to a model, which we use to create the four scenarios under which we perform our analyses.

**Definition 1.** *Unawareness of Sensitive Attribute (Kusner et al. (2017)) The sensitive attribute is not a feature in the space of feature vectors.*

**Definition 2.** *Decorrelation with the Sensitive Attribute (Zemel et al. (2013)) The space of feature vectors satisfies the following property: $P(X|A = a) = P(X|A = b)$.*

## 3 Framework for Analysis

In this section, we present the framework used to analyze the tradeoffs between accuracy and fairness under four scenarios. An illustration of the first stage of our framework is provided in Fig.1a. We construct a discrete approximation, $(\tilde{X}, \tilde{A}, \tilde{Y})$, of $(X, A, Y)$ by first using a generator, $G$, that has learned the latent structure of $(X, A, Y)$ from a dataset to densely sample this distribution. We then partition the feature space into $N_c$ non-intersecting cells $\{C_i\}_1^{N_c}$ that cover $\mathcal{X}$ using vector quantization (VQ) from data compression and signal processing (Linde et al. (1980); Lloyd (1982)). The support of $\tilde{X}$ is given by $\{\mathbf{x}_i^c | \mathbf{x}_i^c \in \mathcal{X}\}_{i=1}^{N_c}$, where $\mathbf{x}_i^c$ represent the centroid of $C_i$. We infer the population statistics of a given cell from the samples inside of it. By densely sampling $G$, we ensure that $(X \in C_j, A, Y) \approx (\tilde{X} = \mathbf{x}_j^c, \tilde{A}, \tilde{Y})$ (see Appendix A for details). For notational simplicity, we use $(X, A, Y)$ in place of $(\tilde{X}, \tilde{A}, \tilde{Y})$ in the remainder of this paper.

An illustration of the different scenarios under which we analyze the fairness-accuracy tradeoff is provided in Fig.1b. The first situation is the unconstrained situation in which we may make direct use of the sensitive attribute to separately assign scores to the feature vectors of different groups (see Appendix F for this formulation). The second situation is formulated in Section 3.1, under which the sensitive attribute is unavailable, meaning that the Bayesian oracle must assign the same score to individuals from different groups with the same feature vectors. When the sensitive attribute is required to be decorrelated with the feature vectors used for classification, an added layer of processing is required to decorrelate the feature vectors from the sensitive attribute prior to providing them to the Bayesian oracle. In the third situation, access to the sensitive attribute allows us to construct two separate mappings, $\mathbf{T}_a$ and $\mathbf{T}_b$, that redistribute the feature vectors associated with each group to achieve this goal (see Appendix G for this formulation). In the fourth situation, the sensitive attribute is unavailable, meaning that a single mapping, $\mathbf{T}$, must be applied to the features of both groups to achieve this goal. This situation is formulated in Section 3.2.

**Consolidating Notation.** To ease notation, we introduce matrix–vector notation that will be used in the ensuing sections for modeling purposes. Bold face capital letters represent matrices, e.g. $\mathbf{X}$, where the value of the $i^{th}$ row and $j^{th}$ column is given by $\mathbf{X}[i, j]$. Bold face lower case letters represent column vectors, e.g. $\mathbf{x}$, where the $i^{th}$ element is given by $\mathbf{x}[i]$. Since $\mathcal{A} = \{a, b\}$ and $\mathcal{Y} = \{0, 1\}$, subscripts (subscripts) containing letters (numbers) refer to joint (conditional) distributions with the sensitive attribute (class label). $\mathbf{p}$ is used to capture a joint distribution with $X$ and other variables, while $\mathbf{q}$ is used to capture a distribution conditioned on $X$. The following examples illustrate this notation. The $i^{th}$ element of the vectors $\mathbf{p}_a$, $\mathbf{p}_0, \mathbf{p}_0^a$, and $\mathbf{q}_a^0$ is equal to $P(X = \mathbf{x}_i^c | A = a)$, $P(X = \mathbf{x}_i^c | Y = 0)$, $P(X = \mathbf{x}_i^c, A = a | Y = 0)$, and $P(Y = 0 | X = \mathbf{x}_i^c, A = a)$, respectively. We use $\mathbf{1}_k$ and $\mathbf{0}_k$ to represent column vectors of length $k$, containing all 1s and all 0s, respectively. $\mathbf{I}_M$ represents an $M \times M$ identity matrix. Finally, $\mathbf{O}_{M,N}$ and $\mathbf{1}_{M,N}$ represent matrices of all zeros and ones with row and column dimensions given by $M$ and $N$, respectively. A table containing all notation introduced in this paper is provided in Appendix M.

### 3.1 Fairness-Accuracy Tradeoff

Given access to the joint distribution $(X, Y)$, the Bayesian oracle takes the majority vote over the support of $X$ and produces the most accurate solution. This solution is given by $\mathbf{s}^B$, where

$$\mathbf{s}^B[i] = \operatorname*{argmax}_y \mathbf{p}^y[i], \forall i. \tag{1}$$

This solution is a special case of a randomized classifier where the outputs are all binary, and thus deterministic. Its accuracy in terms of the probability of correct prediction is given by $Acc_b = \sum_{i=1}^{N_c} \mathbf{p}^{\mathbf{s}^B[i]}[i]$. Let $N_1$ represent the number of cells classified as 1 by the Bayesian classifier and assume, without loss of generality, that these correspond with the first $N_1$ elements of $\mathbf{s}^B$. That is, $\mathbf{s}^B = [\mathbf{1}_{N_1}^T \quad \mathbf{0}_{N_c - N_1}^T]^T$. Then, our goal becomes finding a classifier, $\mathbf{s}^F$, with maximal accuracy that satisfies a set of fairness constraints by minimizing its deviate from $\mathbf{s}^B$. Towards formalizing a minimization problem, let this deviation be

represented by a vector $\mathbf{m}$, given by:

$$\left[\left|\mathbf{s}^B[1] - \mathbf{s}^F[1]\right|, ..., \left|\mathbf{s}^B[N_1] - \mathbf{s}^F[N_1]\right|, -\left|\mathbf{s}^B[N_1 + 1] - \mathbf{s}^F[N_1 + 1]\right|, ..., -\left|\mathbf{s}^B[N_c] - \mathbf{s}^F[N_c]\right|\right]^T. \tag{2}$$

Then, the reduction in accuracy incurred by deviating $\mathbf{s}^F$'s scores from $\mathbf{s}^B$'s scores is given by $(\mathbf{p}^1 - \mathbf{p}^0)^T \mathbf{m}$. Since satisfying a particular notion of fairness exactly may be too strict for a variety of applications, we formulate all fairness constraints as inequalities and provide limits on the degree to which the fair classifier may deviate from exactly satisfying a particular notion of fairness. The list of constraints for each group fairness notion is provided below.

$$|(\mathbf{p}_a - \mathbf{p}_b)^T (\mathbf{s}^B - \mathbf{m})| \le \epsilon_{DP} \quad (DP) \qquad \epsilon_{EOp} = \epsilon_{PE} \tag{EOd}$$

$$|(\mathbf{p}_{a,0} - \mathbf{p}_{b,0})^T (\mathbf{s}^B - \mathbf{m})| \le \epsilon_{EOp} \quad (PE) \qquad |(\mathbf{p}_a^0 - \mathbf{p}_b^0)^T (\mathbf{1}_{N_c} - \mathbf{s}^B + \mathbf{m})$$

$$|(\mathbf{p}_{a,1} - \mathbf{p}_{b,1})^T (\mathbf{s}^B - \mathbf{m})| \le \epsilon_{PE} \quad (EOp) \qquad + (\mathbf{p}_a^1 - \mathbf{p}_b^1)^T (\mathbf{s}^B - \mathbf{m})| \le \epsilon_{EA} \quad (EA) \tag{3}$$

The set of local individual fairness constraints can be formulated as follows:

$$|\mathbf{W}(\mathbf{s}^B - \mathbf{m})| \le \epsilon_{IF} \mathbf{1}_{N_c}, \quad (Ind.) \tag{4}$$

where $\mathbf{W} \in [0,1]^{N_{nbr} \times N_c}$ is a matrix in which the number of rows, $N_{nbr}$, is equal to the total number of feature vector pairs on the support of $X$ within an $\eta$-neighborhood of each other. In particular, the $k^{th}$ row of $\mathbf{W}$ contains non-zero entries, $e^{-\theta d_{\mathcal{X}}^2(\mathbf{x}_i^c, \mathbf{x}_j^c)}$ and $-e^{-\theta d_{\mathcal{X}}^2(\mathbf{x}_i^c, \mathbf{x}_j^c)}$ in only two indices, $i$ and $j$, respectively, for which $d_{\mathcal{X}}(\mathbf{x}_i^c, \mathbf{x}_j^c) \le \eta$. This provides us with the final ingredient required to construct an optimization problem to analyze the fairness-accuracy tradeoff:

$$\min_{\mathbf{m}} (\mathbf{p}^1 - \mathbf{p}^0)^T \mathbf{m}, \quad \text{s.t.} \quad \begin{array}{l} (3) \text{ and } (4) \text{ are satisfied} \\ 0 \le \mathbf{m}[i] \le 1, \qquad 0 \le i \le N_1 \\ -1 \le \mathbf{m}[i] \le 0, \quad N_1 \le i \le N_c \end{array} \tag{5}$$

Observing that each of the constraints in this minimization problem can be made linear in $\mathbf{m}$, this optimization problem is convex and can be efficiently solved using linear programming (Dantzig (1963)). Thus, $\mathbf{s}^F = \mathbf{s}^B - \mathbf{m}$ and the reduction in accuracy is given by $Acc_f = Acc_b - (\mathbf{p}^1 - \mathbf{p}^0)^T \mathbf{m}$. See Appendix C for the explicit derivation of problem (5).

## 3.2 TRANSFER FAIRNESS TO DECORRELATED DOMAIN

Definition 2 measures differences in the distribution of the feature vectors for different groups. Unless the original space of feature vectors is decorrelated with respect to the sensitive attribute, a transformation must be applied to $X$ to satisfy this definition. Specifically, given $(X, A, Y)$, a mapping $T : \mathcal{X} \to \mathcal{X}$ must be constructed to ensure that $P(T(X)|A = a) = P(T(X)|A = b)$. Moreover, we require such a transformation to produce feature vectors which our original fair Bayesian oracle will still fairly classify. That is, $\mathbf{s}^F$ must still be fair with respect to the joint distribution $(T(X), A, Y)$. Since $(X, A, Y)$ is discrete in our framework, this transformation comes in the form of a mixing matrix, $\mathbf{T}$, designed to merge different areas over the support of $X$ without adding or losing information. As a result, we require that $\mathbf{T} \in \mathcal{P}^{N_c \times N_c}$, where $\mathcal{P}^{N_c \times N_c}$ represents the set of $N_c \times N_c$ matrices whose columns are probability mass functions. Hence, $\mathbf{T}$ is a stochastic matrix, the $i^{th}$ column of which determines how the information in the $i^{th}$ VQ cell is disbursed in the transform space. A constraint on decorrelation can be easily constructed by minimizing the value of $\|\mathbf{T}(\mathbf{p_a} - \mathbf{p_b})\|_1$. Hence, the following optimization problem is used to achieve our goal.

$$\min_{\mathbf{T} \in \mathcal{P}^{N_c \times N_c}} -\lambda \underbrace{(\mathbf{s}^{FT} \mathbf{T} \mathbf{p}^1 + (\mathbf{1}_{N_c} - \mathbf{s}^F)^T \mathbf{T} \mathbf{p}^0)}_{Acc_d} + \beta \underbrace{\|\mathbf{T}(\mathbf{p}_a - \mathbf{p}_b)\|_1}_{L_d}$$

$$\text{s.t.} \quad |f(\mathbf{T})| \le \mathbf{f} \quad (Fairness) \tag{6}$$

$Acc_d$ preserves the accuracy of scores produced from applying the fair Bayesian classifier to the transformed space of feature vectors. $L_d$ encourages the transformation to decorrelate

the feature vectors from the sensitive attribute. It can take on a value between 0 (complete decorrelation) and 2 (complete correlation). The *Fairness* constraint directly ensures that the fairness constraints from equations (3) and (4) are preserved after the space of feature vectors has been transformed. $f(\mathbf{T}) \in \mathbb{R}^{(N_{Nbr}+4) \times 1}$ is given by the following equation:

$$
f(\mathbf{T}) = \underbrace{\begin{bmatrix} \mathbf{p_a} - \mathbf{p_b} & \mathbf{0}_{N_c} \\ \mathbf{p_{a1}} - \mathbf{p_{b1}} & \mathbf{0}_{N_c} \\ \mathbf{0}_{N_c} & \mathbf{p_{a0}} - \mathbf{p_{b0}} \\ \mathbf{p_a^1} - \mathbf{p_b^1} & \mathbf{p_a^0} - \mathbf{p_b^0} \\ \mathbf{W} & \mathbf{O}_{N_{Nbr},N_c} \end{bmatrix}}_{\mathbf{P}} \underbrace{\begin{bmatrix} \mathbf{T} & \mathbf{O}_{N_c,N_c} \\ \mathbf{O}_{N_c,N_c} & \mathbf{T} \end{bmatrix}}_{\tilde{\mathbf{T}}} \underbrace{\begin{bmatrix} \mathbf{s}^F \\ (\mathbf{1}_{N_c} - \mathbf{s}^F) \end{bmatrix}}_{\tilde{\mathbf{s}}^F}, \tag{7}
$$

where $\tilde{\mathbf{T}}$ can be directly written as a function of $\mathbf{T}$:

$$
\tilde{\mathbf{T}} = \underbrace{\begin{bmatrix} \mathbf{1}_{N_c \times N_c} & \mathbf{O}_{N_c \times N_c} \\ \mathbf{O}_{N_c \times N_c} & \mathbf{1}_{N_c \times N_c} \end{bmatrix}}_{\mathbf{M}} \circ \left( \underbrace{\begin{bmatrix} \mathbf{I}_{N_c} \\ \mathbf{I}_{N_c} \end{bmatrix}}_{\tilde{\mathbf{I}}} \mathbf{T} \underbrace{[\mathbf{I}_{N_c} \quad \mathbf{I}_{N_c}]}_{\tilde{\mathbf{I}}^T} \right). \tag{8}
$$

The first four elements of $|f(\mathbf{T})|$ capture the degree to which a particular group fairness notion is violated in the transformed space, while the remaining elements capture the degree to which a pair of neighboring feature vectors from the input space violate local individual fairness when transformed (see Appendix D for derivation). Thus, setting $\mathbf{f} = [\epsilon_{DP}, \ \epsilon_{PE}, \ \epsilon_{EOp}, \ \epsilon_{EOd}, \ \mathbf{1}_{N_{nbr}}^T \epsilon_{IF}]^T$ preserves the group and individual fairness constraints (3) and (4).

Note that the (*Fairness*) constraint can be reformulated as an equality constraint, as given by $\max(\tilde{f}(\mathbf{T}) - \tilde{\mathbf{f}}, \mathbf{0}_{2(N_{Nbr}+4)}) = \mathbf{0}_{2(N_{Nbr}+4)}$, where

$$
\tilde{f}(\mathbf{T}) = \begin{bmatrix} -\mathbf{P} \\ \mathbf{P} \end{bmatrix} \tilde{\mathbf{T}} \tilde{\mathbf{s}}^F \qquad \text{and} \qquad \tilde{\mathbf{f}} = \begin{bmatrix} \mathbf{f} \\ \mathbf{f} \end{bmatrix}. \tag{9}
$$

Thus, the question becomes: Given that we must satisfy individual and group fairness constraints associated with the original fair Bayesian oracles's decision map, $\mathbf{s}^F$, how well can we accurately decorrelate the space of feature vectors from the sensitive attribute? To solve this problem, we form the Augmented Lagrangian:

$$
\max_{\boldsymbol{\rho}} \min_{\mathbf{T} \in \mathcal{P}^{N_c \times N_c}} - \lambda(\mathbf{s}^{FT}\mathbf{T}\mathbf{p}^1 + (\mathbf{1}_{N_c} - \mathbf{s}^F)^T \mathbf{T}\mathbf{p}^0) + \beta\|\mathbf{T}(\mathbf{p}_a - \mathbf{p}_b)\|_1
$$
$$
+ \langle \boldsymbol{\rho}, \max(\tilde{f}(\mathbf{T}) - \tilde{\mathbf{f}}, \mathbf{0}_{2(N_{Nbr}+4)}) \rangle + \frac{\tau}{2}\| \max(\tilde{f}(\mathbf{T}) - \tilde{\mathbf{f}}, \mathbf{0}_{2(N_{Nbr}+4)})\|_2^2. \tag{10}
$$

Solving this minimization problem is equivalent to solving minimization problem (6). Thus, if minimization problem (10) is convex, then we can exactly solve, providing us with the solution to minimization problem (6). Thus, we state the following claim before providing a solution to this problem.

**Claim 1.** *Minimization problem (10) is convex.*

*Proof.* See Appendix E. □

Let $L(\mathbf{T}, \boldsymbol{\rho})$ represent the objective function in minimization problem (10). Since this problem is convex, we are able to solve for $\mathbf{T}$ by applying the method of multipliers (Boyd & Vandenberghe (2004)) with the following updates until convergence.

$$
\mathbf{T}^{k+1} = \operatorname*{argmin}_{\mathbf{T} \in \mathcal{P}^{N_c \times N_c}} L(\mathbf{T}, \boldsymbol{\rho}^k) \qquad \text{and} \qquad \boldsymbol{\rho}^{k+1} = \boldsymbol{\rho}^k + \tau \max(\tilde{f}(\mathbf{T}^{k+1}) - \tilde{\mathbf{f}}, \mathbf{0}_{2(N_{Nbr}+4)})
$$
(11)

Optimizing for $\mathbf{T}^k$ in each iteration can be done using a projected subgradient method, where the projection of $\mathbf{T}$ onto $\mathcal{P}^{N_c \times N_c}$ is performed by projecting each column of $\mathbf{T}$ onto the unit simplex (Duchi et al. (2008)). This problem can be generalized to the situation in which the sensitive attribute is allowed to be used to perform the transformation prior to being redacted for classification. In this scenario, we solve for two transformation matrices—one for each group. This problem is still convex and can be solved using the alternating direction method of multipliers algorithm (Boyd & Vandenberghe (2004)) (See Appendix G)

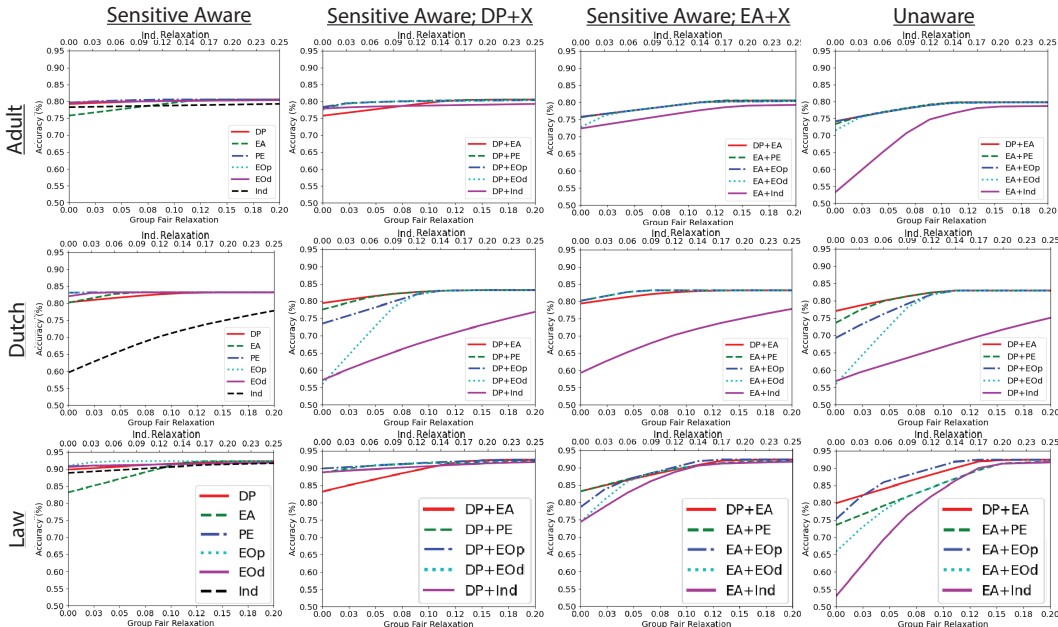

Figure 2: Pareto frontiers capturing the accuracy-fairness tradeoff for three datesets under awareness and unawareness of the sensitive attribute. Each plot provides curves for different pairings of fairness constraints; namely, DP, EA, PE, EOd, and Ind.

## 4 EXPERIMENTAL RESULTS

In this section we experimentally investigate the different modules described in the analysis of this paper on the Adult (Kohavi et al. (1996)), Law (Wightman (1998)), and Dutch Census (Van der Laan (2001)) datasets. Our goals are to analyze the tradeoff between fairness and accuracy when decorrelation is and is not a requirement. Each situation is explored under awareness and unawareness of the sensitive attribute. For more details related to our experimental setups, see Appendix H. For details on how we construct the discrete approximations of the population distribution, see Appendix J. For details on the time complexities associated with our experiments, see Appendix L

### 4.1 ACCURACY-FAIRNESS TRADEOFF

In this section, we analyze the fairness-accuracy tradeoff when feature vector decorrelation from the sensitive attribute is not required. We particularly explore various configurations of minimization problem (5) under both awareness and unawareness of the sensitive attribute.

Fig. 2 provides a panel of Pareto frontiers that summarize the accuracy-fairness tradeoff for different pairings of fairness definitions (see Appendix K for more analyzed combinations). The plots in each row correspond to one of the three datasets. The first three columns provide results under awareness of the sensitive attribute, while the results in the final column are under unawareness of the sensitive attribute. Along the $x$-axis we relax a group fairness constraint (or pair of constraints). The Individual fairness relaxation budget is scaled differently and shown at the top of each plot along the $x$-axis. The $y$-axis of each plot shows the resulting accuracy of the Bayesian oracle operating under the corresponding fairness relaxation budget. Each curve in a plot corresponds to a different constraint setting. For example, a group fairness relaxation of 0.3 for a DP+EA curve means that $\epsilon_{DP} = \epsilon_{EA} = 0.3$, while group fairness relaxation of 0.10 and individual fairness relaxation of 0.12 for a DP+Ind curve means that $\epsilon_{DP} = 0.10$ and $\epsilon_{Ind} = 0.12$. Typically, most group fairness notions can be satisfied exactly in isolation with little accuracy dropoff, as evidenced by the first column of Pareto frontiers. This tends to change when two fairness constraints are paired together. The second column couples DP with each of the other fairness notions, while the third column couples EA with each of the other fairness notions. For the Law and Adult datasets, a tension can be observed in pairings with EA, while pairings with DP are more easily satisfied.

However, the converse is true for the Dutch Census dataset, suggesting that the tension between different group fairness notions is distributionally dependent. The Pareto frontiers in the fourth column of plots are under unawareness of the sensitive attribute. Compared to the situation of awareness, there is a clear deterioration in performance in the Bayesian oracles's accuracy, but the extent of the accuracy dropoff is also distributionally dependent. For example, strictly satisfying EA+PE causes an accuracy reduction of 3% between the awareness and unawareness situations for the Adult dataset. However, for the Law dataset strictly satisfying EA+PE leads to an accuracy dropoff of 10% between the awareness and unawareness situations.

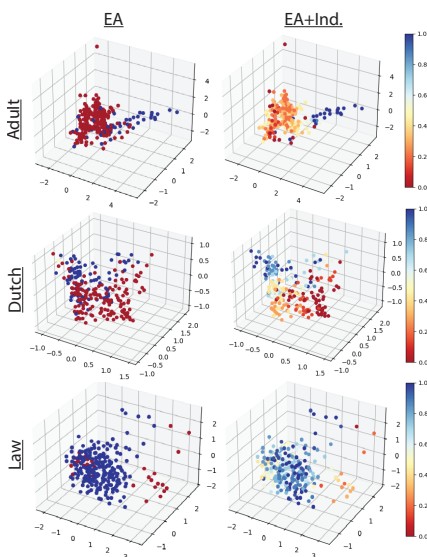

Figure 3: Dimensionality reduction of feature vectors.

Fig. 2 also shows that pairing individual fairness with just one group fairness constraint has the potential to considerably reduce the Bayesian oracles's ability to make fair and accurate decisions, particularly in the case of unawareness. Fig. 3 provides insight into this phenomenon. In this figure, we produce dimensionality reduction plots using factor analysis of mixed data (FAMD), to reduce the dimensions of each VQ cell centroid to three (Pagès (2014)). Each 3D point is color coded according to its score, with points in close proximity representing neighboring VQ cells. The left column of plots displays the results for which EA is exactly satisfied, but without any restrictions on individual fairness. The right column of plots also satisfy EA, but with an imposed individual fairness budget of $\eta = 0.15$. In each of the plots in the left column, collections of red and blue points in close proximity to each other can be observed, indicating that VQ cells in close proximity to each other are receiving drastically different scores. The right column of plots displays a smoother transition of scores, prohibiting the Bayesian oracle from arbitrarily penalizing different VQ cells to satisfy group fairness.

### 4.2 Transfer Fairness to Decorrelated Domain

Table 1: Results for transferring fairness to decorrelated domain for Adult dataset.

| | Awareness | | | | Unawareness | | | |
| --- | --- | --- | --- | --- | --- | --- | --- | --- |
| Fairness Measure | Acc. Reduction | | $L_d$ | | Acc. Reduction | | $L_d$ | |
| DP+EA | 0.005 | (0.008) | 0.000 | (0.000) | 0.007 | (0.008) | 0.000 | (0.000) |
| DP+EOd | 0.008 | (0.007) | 0.000 | (0.000) | 0.010 | (0.007) | 0.000 | (0.000) |
| EA+EOd | 0.010 | (0.012) | 0.061 | (0.084) | 0.012 | (0.005) | 0.050 | (0.080) |
| DP+Ind. | 0.014 | (0.008) | 0.000 | (0.000) | 0.003 | (0.002) | 0.000 | (0.000) |
| EA+Ind. | 0.021 | (0.006) | 0.026 | (0.044) | 0.003 | (0.000) | 0.000 | (0.000) |
| EOd+Ind. | 0.015 | (0.004) | 0.000 | (0.000) | 0.003 | (0.002) | 0.002 | (0.003) |

In this section, we analyze the extent to which the space of feature vectors can be decorrelated with respect to the sensitive attribute while preserving the fairness of the Bayesian oracles's decisions. In our experiments, we set the hyperparameters in minimization problem (10) to $\lambda = 15$ and $\beta = 25$. Table 1 displays the results obtained from our decorrelation analysis under the preservation of different combinations of fairness definitions for the Adult dataset (see Appendix I for results from Law and Dutch Census datasets).

For all combinations involving individual fairness, we hold $\epsilon_{Ind} = 0.05$, meaning that the deviation in the probability of neighboring feature vectors being assigned a positive class label by the Bayesian oracle should be no more than approximately 5%. All group fairness constraints are tested for relaxations of $0.0, 0.5$, and $0.10$. Thus, we report the average over three values in each of the cells of this table along with their standard deviation in parentheses. Prior to applying the decorrelation mapping, the values of $L_d$ under awareness and unawareness were 2 and 0.92, respectively. The results in this table suggest that it is possible to decorrelate the space of feature vectors with little accuracy drop-off in situations in which the sensitive attribute is and is not available for constructing the decorrelation mapping.

That is, in most cases, the accuracy of the fair Bayesian classifier on the decorrelated feature vectors falls by less than 2% on average, with $L_d$ dropping close to 0 in all cases except for the EA+EOd pairing. These results are consistent in the Law and Dutch Census datasets, though these datasets struggle much more to achieve decorrelation for the EA+EOd pairing.

We illustrate the effect of these decorrelation mappings under awareness and unawareness of the sensitive attribute in Fig. 4, plotting the conditional distributions $P(X|A = \text{White})$ and $P(X|A = \text{Non-White})$ over all VQ cells for the Law dataset. The left and right columns of plots provide results under the assumptions of awareness and unawareness of the sensitive attribute, respectively. Under awareness, the number of cells along the $x$-axis doubles compared to unawareness since access to the sensitive attribute allows the Bayesian oracle to separate each cell into two decision regions. Nevertheless, in both cases, the $L_d$ constraint requires all subregions of the space of feature vectors to be

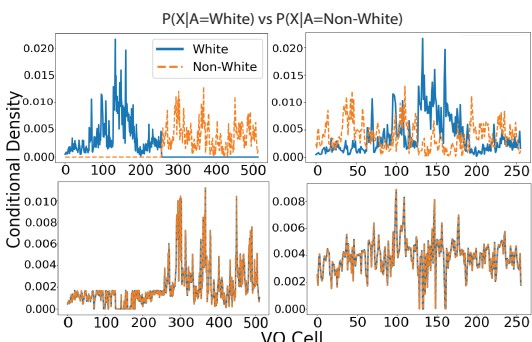

Figure 4: Visualization of non-decorrelated (top) and decorrelated (bottom) feature vectors when the sensitive attribute is (left) and is not (right) used for decorrelation.

decorrelated with respect to the sensitive attribute. These plots demonstrate the effectiveness of the decorrelation mappings with the density of feature vectors in each VQ cell perfectly overlapped in the awareness and unawareness plots in the bottom row. This suggests that the accuracy drop-off of a fair classifier is less dependent on how correlated the features are with the sensitive attribute and more dependent on the strictness of the fairness enforcement.

## 5 Conclusion, Broader Impact, and Limitations

This paper explores the tradeoff between fairness and accuracy under four practical scenarios that limit the data available for classification. We investigate the behavior of a fair Bayesian oracle by approximating the joint distribution of the feature vector, sensitive attribute, and class label. Our exploration encompasses situations in which the sensitive attribute may or may not be available and correlations between feature vectors and the sensitive attribute may or may not be eliminated. Our results also suggest that the pursuit of individual fairness through the enforcement of local scoring consistency may clash with notions of group fairness, particularly when the sensitive attribute is unavailable to a model. Additionally, we demonstrate that it is often feasible to reduce the correlation between the space of feature vectors and the sensitive attribute while preserving the accuracy of a fair model.

**Broader Impact** Fairness in machine learning is a problem of increasing relevance in today's society given the huge increase in applications that use such technology for decision-making. Feasibility studies like ours, which analyze the fairness of such models for real world application scenarios, thus become critical for developing ethical technology. Our study can help developers of fair ML models save time and resources by providing them with a useful tool to understand the extent to which it is even possible for them to produce such models.

**Limitations** We would like to mention two limitations with regards to our approach. First, a sizeable enough dataset for training the generator is needed to model the population distribution of the feature vector, sensitive attribute, and class label. This is, however, not a problem unique to our approach, but relevent to all machine learning applications that rely on the substance of the data used for training. Our experimentation has shown that a dataset of approximately 20,000 samples suffices to model this distribution effectively. Second, quantifying the similarity among individuals introduces subjectivity. The choice of distance metric and the specific parameters used in the metric inherently imply certain assumptions about the similarity between two individuals. Thus, adjusting the distance metric and parameters may produce different results. Notably, this issue is not particular to our framework, but rather a broader challenge inherent in the analysis of individual fairness.

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
