SUPPLEMENTAL MATERIAL

A   MODELING THE POPULATION DISTRIBUTION

Let $X, A$, and $Y$ represent the random feature vector and sensitive attribute and class label random variables respectively associated with the sample spaces $\mathcal{X}, \mathcal{A}$, and $\mathcal{Y}$. For simplicity, we assume that $\mathcal{X} = \mathbb{R}^k, \mathcal{A} = \{a, b\}$, and $\mathcal{Y} = \{0, 1\}$. Let $D_{tr} = \{\mathbf{x}_i^{(tr)}, g_i^{(tr)}, y_i^{(tr)} | \mathbf{x}_i^{(tr)} \in \mathcal{X}, g_i^{(tr)} \in \mathcal{A}, y_i^{(tr)} \in \mathcal{Y}\}_{i=1}^{N_{tr}}$ represent a *dataset*, containing $N_{tr}$ samples from the joint distribution $(X, A, Y)$. A *generator*, $G : \mathcal{L} \to \mathcal{X} \times \mathcal{Y} \times \mathcal{A}$, learns the latent representation of $(X, A, Y)$ from $D_{tr}$ and can be used to sample the distribution $(X, A, Y)$. Let $D = \{\mathbf{x}_i, g_i, y_i | \mathbf{x}_i \in \mathcal{X}, g_i \in \mathcal{A}, y_i \in \mathcal{Y}\}_{i=1}^{N}$ be a dataset produced by generating $N$ samples using $G$.

**Definition 3.** *(Cell decomposition) A cell decomposition of size $N_c$ over $\mathcal{X}$ is given by a disjoint set of $N_c$ cells, $\mathcal{C}_1, ..., \mathcal{C}_{N_c} \subseteq \mathcal{X}$, that cover $\mathcal{X}$. These cells are defined by their centroids, $\mathbf{x}_1^c, ..., \mathbf{x}_{N_c}^c \in \mathcal{X}$, where for each $i \in \mathbb{N}_{\leq N_c}^+$ and for some distance metric, $d_{\mathcal{X}}$:*

$$\mathcal{C}_i = \{\mathbf{x} | \mathbf{x} \in \mathcal{X}, d_{\mathcal{X}}(\mathbf{x}_i^c, \mathbf{x}) \leq d_{\mathcal{X}}(\mathbf{x}_j^c, \mathbf{x}), \forall j \in \mathbb{N}_{\leq N_c}^+ \setminus i\}.$$

We construct a discrete approximation of $(X, A, Y)$, represented by $(\tilde{X}, \tilde{A}, \tilde{Y})$, by inducing a cell decomposition over $\mathcal{X}$, and using $D$ to quantify the statistical properties of $(\tilde{X}, \tilde{A}, \tilde{Y})$ over each cell. Specifically, let $\mathcal{C} = \{\mathbf{x}_i^c | \mathbf{x}_i^c \in \mathcal{X}\}_{i=1}^{N_c}$ be the set containing the $N_c$ centroids produced by a cell decomposition. We construct the probability mass function (pmf) of $\tilde{X}$ as follows:

$$p_{\tilde{X}}(\mathbf{x}) = \begin{cases} \frac{N_i}{N} & \text{, if } \exists i \in \mathbb{N}_{\leq N_c}^+ \text{ s.t. } \mathbf{x} = \mathbf{x}_i^c \\ 0 & \text{, otherwise} \end{cases}, \tag{12}$$

where $N_i$ represents the number of feature vectors from the dataset $D$ in cell $\mathcal{C}_i$. The conditional joint distribution, $\tilde{Y}, \tilde{A} | \tilde{X} = \mathbf{x}_i^c$, is given by:

$$p_{\tilde{A}, \tilde{Y} | \tilde{X} = \mathbf{x}_i^c}(g, y) = \frac{\sum_{j=1}^{N} I[\mathbf{x}_j \in \mathcal{C}_i, g_j = g, y_j = y]}{N_i}, \tag{13}$$

i.e. the portion of samples from $D$ with feature vectors in cell $C_i$ for which $g$ and $y$ are their sensitive attribute and class label. Thus, equations (12) and (13) provide us with the joint distribution $(\tilde{X}, \tilde{A}, \tilde{Y})$.

Since all of our analyses are performed on the approximated distribution, we must sample $G$ densely enough to guarantee that the statistical information inside of each cell of this distribution accurately characterizes the information in each cell of the true population distribution. The following observation provides us with a guide for sampling $G$.

**Observation 1.** *(PAC Bound for sampling generator) Suppose we induce a cell decomposition of size $N_c$ over $\mathcal{X}$ using Lloyd's algorithm for vector quantization (Lloyd (1982); Linde et al. (1980)). Let $j$ refer to an arbitrary cell index. Define $\nu_{j,g}^y = P(A = g, Y = y | X \in C_j)$ and $\mu_{j,g}^y = P(\tilde{A} = g, \tilde{Y} = y | \tilde{X} = \mathbf{x}_j^c)$. Then, for some $\Delta$ and $\delta$, using a total of $N = \frac{-N_c}{2\Delta^2} ln(\frac{1-\delta}{8})$ samples from $G$ will guarantee that $P(\cup_{g \in \mathcal{A}, y \in \mathcal{Y}} |\nu_{j,g}^y - \mu_{j,g}^y| \geq \Delta) \leq 1 - \delta$ on average for all cells.*

*Proof.* Consider a series of 4-dimensional random vectors, $B_1, ..., B_n$, where each random variable in a vector has a Bernoulli distribution characterized by the probability of selecting a feature vector that has class label $Y = y$ and group label $A = g$ or not from cell $\mathcal{C}_j$, where $g \in \mathcal{A}, y \in \mathcal{Y}$. The sample mean and population means of these vectors are given by

$\mu = [\mu_{j,a}^0, \mu_{j,a}^1, \mu_{j,b}^0, \mu_{j,b}^1]^T$ and $\nu = [\nu_{j,a}^0, \nu_{j,a}^1, \nu_{j,b}^0, \nu_{j,b}^1]^T$, respectively. Thus, we have that:

$$P(\cup_{g,y}(|\mu_{j,g}^y - \nu_{j,g}^y| \geq \Delta)) \tag{14}$$

$$\leq \sum_{g,y} P(|\mu_{j,g}^y - \nu_{j,g}^y| \geq \Delta) \text{ ( Union Bound)}$$

$$\leq 8e^{-2\Delta^2 n_j} \text{ (Hoeffding Inequality) },\tag{15}$$

where $n_j$ refers to the number of samples inside cell $\mathcal{C}_j$. Lloyd's algorithm asymptotically induces a cell decomposition in which the density in each cell is equal (Lloyd (1982)). Thus, we assume $n_j = N/N_c$. Now, we can average over (14) and (15) to get:

$$\frac{1}{N_c} \sum_j P(\cup_{g,y}(|\mu_{j,g}^y - \nu_{j,g}^y| \geq \Delta)) \leq \frac{\cancel{N_c}}{\cancel{N_c}} 8e^{-\frac{2\Delta^2 N}{N_c}} \tag{16}$$

Setting $8e^{-\frac{2\Delta^2 N}{N_c}} = 1 - \delta$, and solving for $N$ completes this proof. $\qquad\square$

Thus, Observation 1 provides us with a guide for how densely we should sample $G$ to guarantee the fidelity of the joint distribution of the class label and sensitive attribute over each cell.

## B FAIRNESS DEFINITIONS

**Definition 4.** *Demographic Parity (Kamiran & Calders (2012)) An estimator, $\hat{Y}$, satisfies demographic parity for a binary feature, $A \in \{a, b\}$, if $P(\hat{Y}(X) = 1|A = a) = P(\hat{Y}(X) = 1|A = b)$.*

**Definition 5.** *Equal Accuracy (Zafar et al. (2017)) An estimator, $\hat{Y}$, satisfies equal accuracy for a binary feature, $A \in \{a, b\}$, if $P(\hat{Y}(X) = Y|A = a) = P(\hat{Y}(X) = Y|A = b)$*

**Definition 6.** *Equal Opportunity (Hardt et al. (2016)) An estimator, $\hat{Y}$, satisfies equal opportunity for a binary feature, $A \in \{a, b\}$, if $P(\hat{Y}(X) = 1|A = a, Y = 1) = P(\hat{Y}(X) = 1|A = b, Y = 1)$.*

**Definition 7.** *Predictive Equality (Chouldechova (2017)) An estimator, $\hat{Y}$, satisfies predictive equality for a binary feature, $A \in \{a, b\}$, if $P(\hat{Y}(X) = 1|A = a, Y = 0) = P(\hat{Y}(X) = 1|A = b, Y = 0)$.*

**Definition 8.** *Equalized Odds (Hardt et al. (2016)) An estimator, $\hat{Y}$, satisfies equalized odds if both equal opportunity and predictive equality are satisfied.*

**Definition 9.** *Local Individual Fairness (Petersen et al. (2021)) A scoring function, $S$, is locally individually fair if for $L \geq 0$,*

$$\mathbb{E}_{\mathbf{x}_i \sim P_X} \left[ \limsup_{\mathbf{x}_i : d_{\mathcal{X}}(\mathbf{x}_i, \mathbf{x}_j) \downarrow 0} \frac{d_{\mathcal{Y}}(S(\mathbf{x}_i), S(\mathbf{x}_j))}{d_{\mathcal{X}}(\mathbf{x}_i, \mathbf{x}_j)} \right] \leq L \leq \infty. \tag{17}$$

Note that we use the local definition of individual fairness provided by Petersen et al. (2021) as opposed to the original definition given by Dwork et al. (2012) since that latter contains $\mathcal{O}(N_c^2)$ constraints. The local definition allows us to directly construct a system of constraints, which in practice, is much smaller than the number provided in the original definition by Dwork et al. (2012). Namely, the following constraints are imposed:

$$e^{-\theta d_{\mathcal{X}}^2(\mathbf{x}_i, \mathbf{x}_j)} |s^F[i] - s^F[j]| \leq \epsilon_{IF}, \qquad \forall i, j \quad \text{s.t.} \quad d_{\mathcal{X}}(\mathbf{x}_i, \mathbf{x}_j) \leq \eta, \tag{18}$$

where we only require feature vectors falling within an $\eta$-neighborhood of each to receive similar scores.

## C Derivation of Minimization Problem (5)

Let $S^B : \mathcal{X} \to [0, 1]$ and $S^F : \mathcal{X} \to [0, 1]$ respectively represent the optimal unconstrained and fair randomized scoring functions whose outputs represent the probability of producing a 1 label for a given feature vector. Furthermore, let

$$\hat{Y}^B(\mathbf{x}) = \begin{cases} 1 & , \text{ w.p. } S^B(\mathbf{x}) \\ 0 & , \text{ w.p. } 1 - S^B(\mathbf{x}) \end{cases} \quad \text{and} \quad \hat{Y}^F(\mathbf{x}) = \begin{cases} 1 & , \text{ w.p. } S^F(\mathbf{x}) \\ 0 & , \text{ w.p. } 1 - S^F(\mathbf{x}) \end{cases}$$

With this information, we derive the objective function in minimization problem (5), which represents the reduction in accuracy between the unconstrained and fair Bayesian oracles, by showing that it is equivalent to:

$$P(\hat{Y}^B(X) = Y) - P(\hat{Y}^F(X) = Y)$$

We further derive the group fairness linear constraints provided in equation (5), by showing that they are equivalent the the following list of constraints:

$$|P(\hat{Y}^F(X) = 1|A = a) - P(\hat{Y}^F(X) = 1|A = b)| \leq \epsilon_{DP}$$
$$|P(\hat{Y}^F(X) = 1|A = a, Y = 1) - P(\hat{Y}^F(X) = 1|A = b, Y = 1)| \leq \epsilon_{EOp}$$
$$|P(\hat{Y}^F(X) = 1|A = a, Y = 0) - P(\hat{Y}^F(X) = 1|A = b, Y = 0)| \leq \epsilon_{PE}$$
$$|P(\hat{Y}^F(X) = Y|A = a) - P(\hat{Y}^F(X) = Y|A = b)| \leq \epsilon_{EA},$$

Note that equalized odds is omitted since it is simply a combination of predictive equality and equal opportunity. We also omit the derivation of the individual fairness constraints since they are constructed in a straightforward manner in the body of the paper.

### C.1 Objective Function:

Observe that:

$$P(\hat{Y}^B(X) = Y) - P(\hat{Y}^F(X) = Y) = \sum_{i=1}^{N_c} \left[ P(\hat{Y}^B(X) = Y, X = \mathbf{x}_i^c) - P(\hat{Y}^F(X) = Y, X = \mathbf{x}_i^c) \right]$$

$$= \sum_{i=1}^{N_c} \Big[ P(\hat{Y}^B(X) = 1, Y = 1, X = \mathbf{x}_i^c) + P(\hat{Y}^B(X) = 0, Y = 0, X = \mathbf{x}_i^c)$$
$$- P(\hat{Y}^F(X) = 1, Y = 1, X = \mathbf{x}_i^c) - P(\hat{Y}^F(X) = 0, Y = 0, X = \mathbf{x}_i^c) \Big]$$

$$= \sum_{i=1}^{N_c} \Big[ \{ P(\hat{Y}^B(X) = 1 | \underbrace{Y = 1}_{\substack{\text{Conditional independence given } X. \\ \text{Similar for other cancellations.}}}, X = \mathbf{x}_i^c) - P(\hat{Y}^F(X) = 1 | Y = 1, X = \mathbf{x}_i^c) \} \underbrace{P(Y = 1, X = \mathbf{x}_i^c)}_{\substack{= \mathbf{p}^1[i] \text{ by definition.} \\ \text{Similar for other substitutions.}}}$$
$$+ \{ P(\hat{Y}^B(X) = 0 | Y = 0, X = \mathbf{x}_i^c) - P(\hat{Y}^F(X) = 0 | Y = 0, X = \mathbf{x}_i^c) \} P(Y = 0, X = \mathbf{x}_i^c) \Big]$$

$$= \sum_{i=1}^{N_c} \Big[ \{ \underbrace{P(\hat{Y}^B(X) = 1 | X = \mathbf{x}_i^c)}_{\substack{= \mathbf{s}^B[i] \text{ by definition.} \\ \text{Similar for other substitutions.}}} - P(\hat{Y}^F(X) = 1 | X = \mathbf{x}_i^c) \} \mathbf{p}^1[i]$$
$$+ \{ P(\hat{Y}^B(X) = 0 | X = \mathbf{x}_i^c) - P(\hat{Y}^F(X) = 0 | X = \mathbf{x}_i^c) \} \mathbf{p}^0[i] \Big]$$

$$= \sum_{i=1}^{N_c} \Big[ \{ \mathbf{s}^B[i] - \mathbf{s}^F[i] \} \mathbf{p}^1[i] + \{ 1 - \mathbf{s}^B[i] - 1 + \mathbf{s}^F[i] \} \mathbf{p}^0[i] \Big]$$

$$= \sum_{i=1}^{N_c} \Big[ (\mathbf{s}^B[i] - \mathbf{s}^F[i])(\mathbf{p}^1[i] - \mathbf{p}^0[i]) \Big] \underset{(A)}{=} \sum_{i=1}^{N_c} \mathbf{m}[i](\mathbf{p}^1[i] - \mathbf{p}^0[i]) = \mathbf{m}^T(\mathbf{p}^1 - \mathbf{p}^0)$$

where $(A)$ holds since

$$= \sum_{i=1}^{N_c} \left[ (\mathbf{s}^B[i] - \mathbf{s}^F[i])(\mathbf{p}^1[i] - \mathbf{p}^0[i]) \right]$$

$$= \sum_{i=1}^{N_1} \left[ \underbrace{(\mathbf{s}^B[i] - \mathbf{s}^F[i])}_{\geq 0 \text{ since } \mathbf{s}^B[i] = 1 \text{ for } i \leq N_1}(\mathbf{p}^1[i] - \mathbf{p}^0[i]) \right] + \sum_{i=N_1+1}^{N_c} \left[ \underbrace{(\mathbf{s}^B[i] - \mathbf{s}^F[i])}_{\leq 0 \text{ since } \mathbf{s}^B[i] = 0 \text{ for } i > N_1}(\mathbf{p}^1[i] - \mathbf{p}^0[i]) \right]$$

$$= \sum_{i=1}^{N_1} \left[ \underbrace{|\mathbf{s}^B[i] - \mathbf{s}^F[i]|}_{\mathbf{m}[i] \text{ by definition for } i \leq N_1.}(\mathbf{p}^1[i] - \mathbf{p}^0[i]) \right] + \sum_{i=N_1+1}^{N_c} \left[ \underbrace{-|\mathbf{s}^B[i] - \mathbf{s}^F[i]|}_{\mathbf{m}[i] \text{ by definition for } i > N_1}(\mathbf{p}^1[i] - \mathbf{p}^0[i]) \right]$$

$$= \sum_{i=1}^{N_1} \mathbf{m}[i](\mathbf{p}^1[i] - \mathbf{p}^0[i]) + \sum_{i=N_1+1}^{N_c} \mathbf{m}[i](\mathbf{p}^1[i] - \mathbf{p}^0[i]) = \sum_{i=1}^{N_c} \mathbf{m}[i](\mathbf{p}^1[i] - \mathbf{p}^0[i]).$$

## C.2 DEMOGRAPHIC PARITY:

Observe that:

$$P(\hat{Y}^F(X) = 1 | A = a) = \sum_{i=1}^{N_c} P(\hat{Y}^F(X) = 1, X = \mathbf{x}_i^c | A = a)$$

$$= \sum_{i=1}^{N_1} P(\hat{Y}^F(X) = 1, X = \mathbf{x}_i^c | A = a) + \sum_{i=N_1+1}^{N_c} P(\hat{Y}^F(X) = 1, X = \mathbf{x}_i^c | A = a)$$

$$= \sum_{i=1}^{N_1} \overbrace{P(\hat{Y}^F(X) = 1 | X = \mathbf{x}_i^c, \underbrace{A = a)}_{\substack{\text{Conditional independence given } X. \\ \text{Similar for other cancellations.}}}}^{\mathbf{s}^F[i] = \mathbf{s}^B[i] - (\mathbf{s}^B[i] - \mathbf{s}^F[i]) \text{ by definition.}} P(X = \mathbf{x}_i^c | A = a)$$

$$+ \sum_{i=N_1+1}^{N_c} \underbrace{P(\hat{Y}^F(X) = 1 | X = \mathbf{x}_i^c, A = a)}_{\mathbf{s}^F[i] = \mathbf{s}^B[i] - (\mathbf{s}^B[i] - \mathbf{s}^F[i]) \text{ by definition}} P(X = \mathbf{x}_i^c | A = a)$$

$$= \sum_{i=1}^{N_1} \left[ \mathbf{s}^B[i] - (\mathbf{s}^B[i] - \mathbf{s}^F[i]) \right] \underbrace{P(X = \mathbf{x}_i^c | A = a)}_{\substack{= \mathbf{p}_a[i] \text{ by definition.} \\ \text{Similar for other substitutions.}}} + \sum_{i=N_1+1}^{N_c} \left[ \mathbf{s}^B[i] - (\mathbf{s}^B[i] - \mathbf{s}^F[i]) \right] P(X = \mathbf{x}_i^c | A = a)$$

$$= \Big( \underbrace{\left[ \mathbf{s}^B[1], \quad \dots, \quad \mathbf{s}^B[N_1], \quad \mathbf{s}^B[N_1 + 1], \quad \dots, \quad \mathbf{s}^B[N_c] \right]}_{\mathbf{s}^{BT}}$$

$$- \underbrace{\left[ \underbrace{\mathbf{s}^B[1] - \mathbf{s}^F[1], \quad \dots, \quad \mathbf{s}^B[N_1] - \mathbf{s}^F[N_1]}_{\geq 0 \text{ since } \mathbf{s}^B[i] = 1 \text{ for } i \leq N_1}, \quad \underbrace{\mathbf{s}^B[N_1 + 1] - \mathbf{s}^F[N_1 + 1], \quad \dots, \quad \mathbf{s}^B[N_c] - \mathbf{s}^F[N_c]}_{\leq 0 \text{ since } \mathbf{s}^B[i] = 0 \text{ for } i > N_1} \right]}_{\left[ \left| \mathbf{s}^B[1] - \mathbf{s}^F[1] \right|, \dots, \left| \mathbf{s}^B[N_1] - \mathbf{s}^F[N_1] \right|, -\left| \mathbf{s}^B[N_1 + 1] - \mathbf{s}^F[N_1 + 1] \right|, \dots, -\left| \mathbf{s}^B[N_c] - \mathbf{s}^F[N_c] \right| \right]}_{\mathbf{m}^T} \Big) \mathbf{p}_a$$

$$= (\mathbf{s}^B - \mathbf{m})^T \mathbf{p}_a$$

A similar derivation of $P(\hat{Y}^F(X) = 1 | A = b)$ yields: $P(\hat{Y}^F(X) = 1 | A = b) = (\mathbf{s}^B - \mathbf{m})^T \mathbf{p}_b$.
Thus,

$$|P(\hat{Y}^F(X) = 1 | A = a) - P(\hat{Y}^F(X) = 1 | A = b)| \leq \epsilon_{DP}$$

is equivalent to

$$|(\mathbf{s}^B - \mathbf{m})^T (\mathbf{p}_a - \mathbf{p}_b)| \leq \epsilon_{DP} \qquad \text{or} \qquad |(\mathbf{p}_a - \mathbf{p}_b)^T (\mathbf{s}^B - \mathbf{m})| \leq \epsilon_{DP}.$$

## C.3 EQUAL OPPORTUNITY:

Observe that:

$$P(\hat{Y}^F(X) = 1 | A = a, Y = 1) = \sum_{i=1}^{N_c} P(\hat{Y}^F(X) = 1, X = \mathbf{x}_i^c | A = a, Y = 1)$$

$$= \sum_{i=1}^{N_1} P(\hat{Y}^F(X) = 1, X = \mathbf{x}_i^c | A = a, Y = 1) + \sum_{i=N_1+1}^{N_c} P(\hat{Y}^F(X) = 1, X = \mathbf{x}_i^c | A = a, Y = 1)$$

$$= \sum_{i=1}^{N_1} \overbrace{P(\hat{Y}^F(X) = 1 | X = \mathbf{x}_i^c, \underbrace{A = a, Y = 1}_{\substack{\text{Conditional independence given } X. \\ \text{Similar for other cancellations.}}})}^{\mathbf{s}^F[i] = \mathbf{s}^B[i] - (\mathbf{s}^B[i] - \mathbf{s}^F[i]) \text{ by definition.}} P(X = \mathbf{x}_i^c | A = a, Y = 1)$$

$$+ \sum_{i=N_1+1}^{N_c} \underbrace{P(\hat{Y}^F(X) = 1 | X = \mathbf{x}_i^c, \underbrace{A = a, Y = 1})}_{\mathbf{s}^F[i] = \mathbf{s}^B[i] - (\mathbf{s}^B[i] - \mathbf{s}^F[i]) \text{ by definition.}} P(X = \mathbf{x}_i^c | A = a, Y = 1)$$

$$= \sum_{i=1}^{N_1} \left[ \mathbf{s}^B[i] - (\mathbf{s}^B[i] - \mathbf{s}^F[i]) \right] \underbrace{P(X = \mathbf{x}_i^c | A = a, Y = 1)}_{\substack{= \mathbf{p}_{a,1}[i] \text{ by definition.} \\ \text{Similar for other substitutions.}}}$$

$$+ \sum_{i=N_1+1}^{N_c} \left[ \mathbf{s}^B[i] - (\mathbf{s}^B[i] - \mathbf{s}^F[i]) \right] P(X = \mathbf{x}_i^c | A = a, Y = 1)$$

$$\underbrace{= (\mathbf{s}^B - \mathbf{m})^T \mathbf{p}_{a,1}}_{\text{Similar to the demographic parity derivation.}}$$

A similar derivation of $P(\hat{Y}^F(X) = 1 | A = b, Y = 1)$ yields: $P(\hat{Y}^F(X) = 1 | A = b, Y = 1) = (\mathbf{s}^B - \mathbf{m})^T \mathbf{p}_{b,1}$. Thus,

$$|P(\hat{Y}^F(X) = 1 | A = a, Y = 1) - P(\hat{Y}^F(X) = 1 | A = b, Y = 1)| \le \epsilon_{EOp}$$

is equivalent to

$$|(\mathbf{s}^B - \mathbf{m})^T (\mathbf{p}_{a,1} - \mathbf{p}_{b,1})| \le \epsilon_{EOp} \qquad \text{or} \qquad |(\mathbf{p}_{a,1} - \mathbf{p}_{b,1})^T (\mathbf{s}^B - \mathbf{m})| \le \epsilon_{EOp}.$$

## C.4 PREDICTIVE EQUALITY

This derivation is closely related to the Equal Opportunity derivation. Hence we omit it, directly claiming that

$$|P(\hat{Y}^F(X) = 1 | A = a, Y = 0) - P(\hat{Y}^F(X) = 1 | A = b, Y = 0)| \le \epsilon_{PE}$$

is equivalent to

$$|(\mathbf{p}_{a,0} - \mathbf{p}_{b,0})^T (\mathbf{s}^B - \mathbf{m})| \le \epsilon_{PE}.$$

## C.5 Equal Accuracy:

Observe that:

$$P(\hat{Y}^F(X) = Y|A = a) = P(\hat{Y}^F(X) = 1, Y = 1|A = a) + P(\hat{Y}^F(X) = 0, Y = 0|A = a)$$

$$= \sum_{i=1}^{N_c} P(\hat{Y}^F(X) = 1, Y = 1, X = \mathbf{x}_i^c|A = a) + P(\hat{Y}^F(X) = 0, Y = 0, X = \mathbf{x}_i^c|A = a)$$

$$= \sum_{i=1}^{N_c} P(\hat{Y}^F(X) = 1|X = \mathbf{x}_i^c, \underbrace{A = a, Y = 1}_{\substack{\text{Conditional independence given } X. \\ \text{Similar for other cancellations.}}})P(Y = 1, X = \mathbf{x}_i^c|A = a)$$

$$+ \sum_{i=1}^{N_c} P(\hat{Y}^F(X) = 0|X = \mathbf{x}_i^c, \underbrace{A = a, Y = 0})P(Y = 0, X = \mathbf{x}_i^c|A = a)$$

$$= \sum_{i=1}^{N_1} \overbrace{P(\hat{Y}^F(X) = 1|X = \mathbf{x}_i^c)}^{\mathbf{s}^F[i] = \mathbf{s}^B[i] - (\mathbf{s}^B[i] - \mathbf{s}^F[i]) \text{ by definition.}} \underbrace{P(Y = 1, X = \mathbf{x}_i^c|A = a)}_{\substack{= \mathbf{p}_a^1[i] \text{ by definition.} \\ \text{Similar for other substitutions.}}}$$

$$+ \sum_{i=1}^{N_1} \overbrace{P(\hat{Y}^F(X) = 0|X = \mathbf{x}_i^c)}^{1 - \mathbf{s}^F[i] = 1 - \mathbf{s}^B[i] + (\mathbf{s}^B[i] - \mathbf{s}^F[i]) \text{ by definition.}} P(Y = 0, X = \mathbf{x}_i^c|A = a)$$

$$+ \sum_{i=N_1+1}^{N_c} \overbrace{P(\hat{Y}^F(X) = 1|X = \mathbf{x}_i^c)}^{\mathbf{s}^F[i] = \mathbf{s}^B[i] - (\mathbf{s}^B[i] - \mathbf{s}^F[i]) \text{ by definition.}} P(Y = 1, X = \mathbf{x}_i^c|A = a)$$

$$+ \sum_{i=N_1+1}^{N_c} \overbrace{P(\hat{Y}^F(X) = 0|X = \mathbf{x}_i^c)}^{1 - \mathbf{s}^F[i] = 1 - \mathbf{s}^B[i] + (\mathbf{s}^B[i] - \mathbf{s}^F[i]) \text{ by definition.}} P(Y = 0, X = \mathbf{x}_i^c|A = a)$$

$$= \Big( \underbrace{\big[\mathbf{s}^B[1], \quad \ldots, \quad \mathbf{s}^B[N_1], \quad \mathbf{s}^B[N_1+1], \quad \ldots, \quad \mathbf{s}^B[N_c]\big]}_{\mathbf{s}^{BT}}$$

$$- \underbrace{\Big[\underbrace{\mathbf{s}^B[1] - \mathbf{s}^F[1], \quad \ldots, \quad \mathbf{s}^B[N_1] - \mathbf{s}^F[N_1]}_{\geq 0 \text{ since } \mathbf{s}^B[i] = 1 \text{ for } i \leq N_1}, \quad \underbrace{\mathbf{s}^B[N_1+1] - \mathbf{s}^F[N_1+1], \quad \ldots, \quad \mathbf{s}^B[N_c] - \mathbf{s}^F[N_c]}_{\leq 0 \text{ since } \mathbf{s}^B[i] = 0 \text{ for } i > N_1}\Big]}_{\mathbf{m}^T}\Big)\mathbf{p}_a^1$$

$$+ \Big( \underbrace{\big[1 \quad 1 \quad \ldots \quad 1\big]}_{\mathbf{1}_{N_c}^T} - \underbrace{\big[\mathbf{s}^B[1], \quad \ldots, \quad \mathbf{s}^B[N_c]\big]}_{\mathbf{s}^{BT}}$$

$$+ \underbrace{\Big[\underbrace{\mathbf{s}^B[1] - \mathbf{s}^F[1], \quad \ldots, \quad \mathbf{s}^B[N_1] - \mathbf{s}^F[N_1]}_{\geq 0 \text{ since } \mathbf{s}^B[i] = 1 \text{ for } i \leq N_1}, \quad \underbrace{\mathbf{s}^B[N_1+1] - \mathbf{s}^F[N_1+1], \quad \ldots, \quad \mathbf{s}^B[N_c] - \mathbf{s}^F[N_c]}_{\leq 0 \text{ since } \mathbf{s}^B[i] = 0 \text{ for } i > N_1}\Big]}_{\mathbf{m}^T}\Big)\mathbf{p}_a^0$$

$$= (\mathbf{s}^B - \mathbf{m})^T\mathbf{p}_a^1 + (\mathbf{1}_{N_c} - \mathbf{s}^B + \mathbf{m})^T\mathbf{p}_a^0.$$

Through similar derivation, we obtain $P(\hat{Y}^F(X) = Y|A = b) = (\mathbf{s}^B - \mathbf{m})^T\mathbf{p}_b^1 + (\mathbf{1}_{N_c} - \mathbf{s}^B + \mathbf{m})^T\mathbf{p}_b^0$. Hence,

$$|P(\hat{Y}^F(X) = Y|A = a) - P(\hat{Y}^F(X) = Y|A = b)| \leq \epsilon_{EA}$$

reduces to

$$|(\mathbf{s}^B - \mathbf{m})^T(\mathbf{p}_a^1 - \mathbf{p}_b^1) + (\mathbf{1}_{N_c} - \mathbf{s}^B + \mathbf{m})^T(\mathbf{p}_a^0 - \mathbf{p}_b^0)| \leq \epsilon_{EA}$$

or

$$|(\mathbf{p}_a^1 - \mathbf{p}_b^1)^T(\mathbf{s}^B - \mathbf{m}) + (\mathbf{p}_a^0 - \mathbf{p}_b^0)^T(\mathbf{1}_{N_c} - \mathbf{s}^B + \mathbf{m})| \leq \epsilon_{EA}.$$

## D  Derivation of Minimization Problem (6)

Let $S^F : \mathcal{X} \to [0,1]$ be the optimal fair randomized scoring function whose output represents the probability of producing a 1 label for a given feature vector, and let:

$$\hat{Y}^F(\mathbf{x}) = \begin{cases} 1 & \text{, w.p. } S^F(\mathbf{x}) \\ 0 & \text{, w.p. } 1 - S^F(\mathbf{x}) \end{cases}$$

be the optimal fair classifier for the original non-transformed space of feature vectors. In this section, we derive the objective function and fairness constraints used in minimization problem (6). We will first show that $Acc_d = P(\hat{Y}^F(T(X)) = Y)$, which means that minimizing $-Acc_d$ is equivalent to maximizing the fair Bayesian oracle's accuracy on the space of transformed feature vectors. $L_d$ is self-explanatory, and thus no derivation is required. We will derive each of the fairness constraints separately by showing that they are equivalent to the the following list of constraints (again, equalized odds is omitted since it is simply a combination or predictive equality and equal opportunity):

$$|P(\hat{Y}^F(T(X)) = 1|A = a) - P(\hat{Y}^F(T(X)) = 1|A = b)| \leq \epsilon_{DP}$$
$$|P(\hat{Y}^F(T(X)) = 1|A = a, Y = 1) - P(\hat{Y}^F(T(X)) = 1|A = b, Y = 1)| \leq \epsilon_{EOp}$$
$$|P(\hat{Y}^F(T(X)) = 1|A = a, Y = 0) - P(\hat{Y}^F(T(X)) = 1|A = b, Y = 0)| \leq \epsilon_{PE}$$
$$|P(\hat{Y}^F(T(X)) = Y|A = a) - P(\hat{Y}^F(T(X)) = Y|A = b)| \leq \epsilon_{EA}$$
$$e^{-\theta d_{\mathcal{X}}^2(\mathbf{x}_i, \mathbf{x}_j)}|t^F[i] - t^F[j]| \leq \epsilon_{IF}, \qquad \forall i,j \quad \text{s.t.} \quad d_{\mathcal{X}}(\mathbf{x}_i, \mathbf{x}_j) \leq \eta,$$

where for the individual fairness constraint, $t[i]$ represents the score associated with feature vectors from the original $i^{th}$ VQ cell *after* the decorrelation transformation has been applied. The $(Fairness)$ constraint in minimization problem (6) simply structures all of these constraints in one block matrix form.

### D.1  Derivation of $Acc_d$

Observe that:

$$P(\hat{Y}^F(T(X)) = Y) = P(\hat{Y}^F(T(X)) = 1, Y = 1) + P(\hat{Y}^F(T(X)) = 0, Y = 0)$$
$$= \sum_{i=1}^{N_c} \{ \underbrace{P(\hat{Y}^F(T(X)) = 1, Y = 1, T(X) = \mathbf{x}_i^c)}_{(A)} + \underbrace{P(\hat{Y}^F(T(X)) = 0, Y = 0, T(X) = \mathbf{x}_i^c)}_{(B)} \}$$

$$(19)$$

The derivations to show that terms $(A)$ and $(B)$ are respectively equal $\mathbf{s}^{FT}\mathbf{T}\mathbf{p}^1$ and $(\mathbf{1}_{N_c} - \mathbf{s}^F)^T\mathbf{T}\mathbf{p}^0$ are similar. Thus, for brevity, we only provide the former derivation.

$$P(\hat{Y}^F(T(X)) = 1, Y = 1, T(X) = \mathbf{x}_i^c)$$
$$= \overbrace{P(\hat{Y}^F(T(X)) = 1|T(X) = \mathbf{x}_i^c, \underbrace{Y = 1}_{\text{Conditional independence given } T(X).})}^{\mathbf{s}^F[i] \text{ by definition.}} P(T(X) = \mathbf{x}_i^c, Y = 1)$$

$$(20)$$

Next, note that

$$P(T(X) = \mathbf{x}_i^c, Y = 1) = \sum_{k=1}^{N_c} P(T(X) = \mathbf{x}_i^c, Y = 1, X = \mathbf{x}_k^c)$$
$$= \sum_{k=1}^{N_c} \underbrace{P(T(X) = \mathbf{x}_i^c|X = \mathbf{x}_k^c, \underbrace{Y = 1}_{\text{Conditional independence given } X)}}_{\mathbf{T}[i,k]} \underbrace{P(Y = 1, X = \mathbf{x}_k^c)}_{\mathbf{p}^1[k]} = \sum_{k=1}^{N_c} \mathbf{T}[i,k]\mathbf{p}^1[k]. \qquad (21)$$

Plugging (21) into (20), we have that $(A)$ is given by:

$$P(\hat{Y}^F(T(X)) = 1, Y = 1, T(X) = \mathbf{x}_i^c) = \mathbf{s}^F[i] \sum_{k=1}^{N_c} \mathbf{T}[i,k]\mathbf{p}^1[k]. \tag{22}$$

Through a similar derivation, coupled with the fact that $P(\hat{Y}^F(T(X)) = 0|T(X) = \mathbf{x}_i^c, Y = 0) = 1 - \mathbf{s}^F[i]$, we can find that $(B)$ is given by:

$$P(\hat{Y}^F(T(X)) = 0, Y = 0, T(X) = \mathbf{x}_i^c) = (1 - \mathbf{s}^F[i]) \sum_{k=1}^{N_c} \mathbf{T}[i,k]\mathbf{p}^0[k]. \tag{23}$$

Finally, substituting (22) and (23) into (19), we obtain that

$$P(\hat{Y}^F(T(X)) = Y) = \sum_{i=1}^{N_c} \{ \mathbf{s}^F[i] \sum_{k=1}^{N_c} \mathbf{T}[i,k]\mathbf{p}^1[k] + (1 - \mathbf{s}^F[i]) \sum_{k=1}^{N_c} \mathbf{T}[i,k]\mathbf{p}^0[k] \}$$

$$= \sum_{i=1}^{N_c} \sum_{k=1}^{N_c} \mathbf{T}[i,k]\mathbf{s}^F[i]\mathbf{p}^1[k] + \sum_{i=1}^{N_c} \sum_{k=1}^{N_c} \mathbf{T}[i,k](1 - \mathbf{s}^F[i])\mathbf{p}^0[k]$$

$$= \mathbf{s}^{FT}\mathbf{T}\mathbf{p}^1 + (\mathbf{1}_{N_c} - \mathbf{s}^F)^T\mathbf{T}\mathbf{p}^0 = Acc_d.$$

### D.2 DEMOGRAPHIC PARITY:

Observe that:

$$P(\hat{Y}^F(T(X)) = 1|A = a) = \sum_{i=1}^{N_c} P(\hat{Y}^F(T(X)) = 1, T(X) = \mathbf{x}_i^c|A = a)$$

$$= \sum_{i=1}^{N_c} \overbrace{P(\hat{Y}^F(T(X)) = 1|T(X) = \mathbf{x}_i^c, \underbrace{A = a}_{\text{Conditional independence given } T(X).})}^{\mathbf{s}^F[i] \text{ by definition.}} P(T(X) = \mathbf{x}_i^c|A = a) \tag{24}$$

Next, note that:

$$P(T(X) = \mathbf{x}_i^c|A = a) = \sum_{k=1}^{N_c} P(T(X) = \mathbf{x}_i^c, X = \mathbf{x}_k^c|A = a)$$

$$= \sum_{k=1}^{N_c} P(T(X) = \mathbf{x}_i^c|X = \mathbf{x}_k^c, \underbrace{A = a}_{\text{Conditional independence given } X.})P(X = \mathbf{x}_k^c|A = a) = \sum_{k=1}^{N_c} \mathbf{T}[i,k]\mathbf{p}_a[k]. \tag{25}$$

Plugging (25) into (24), we obtain:

$$P(\hat{Y}^F(T(X)) = 1|A = a) = \sum_{k=1}^{N_c} \mathbf{s}^F[i] \sum_{k=1}^{N_c} \mathbf{T}[i,k]\mathbf{p}_a[k] = \sum_{k=1}^{N_c} \sum_{k=1}^{N_c} \mathbf{T}[i,k]\mathbf{s}^F[i]\mathbf{p}_a[k] = \mathbf{p}_a^T\mathbf{T}\mathbf{s}^F.$$

A similar derivation yields that $P(\hat{Y}^F(T(X)) = 1|A = b) = \mathbf{p}_b^T\mathbf{T}\mathbf{s}^F$. Thus,

$$|P(\hat{Y}^F(T(X)) = 1|A = a) - P(\hat{Y}^F(T(X)) = 1|A = b)| \leq \epsilon_{DP}$$

is equivalent to

$$|(\mathbf{p}_a - \mathbf{p}_b)^T\mathbf{T}\mathbf{s}^F| \leq \epsilon_{DP}.$$

### D.3 EQUAL OPPORTUNITY:

Observe that:

$$P(\hat{Y}^F(T(X)) = 1|A = a, Y = 1) = \sum_{i=1}^{N_c} P(\hat{Y}^F(T(X)) = 1, T(X) = \mathbf{x}_i^c|A = a, Y = 1)$$

$$= \sum_{i=1}^{N_c} \overbrace{P(\hat{Y}^F(T(X)) = 1|T(X) = \mathbf{x}_i^c, \underbrace{A = a, Y = 1}_{\text{conditional independence}})}^{\mathbf{s}^F[i] \text{ by definition.}} P(T(X) = \mathbf{x}_i^c|A = a, Y = 1). \tag{26}$$

Next, note that:

$$P(T(X) = \mathbf{x}_i^c | A = a, Y = 1) = \sum_{k=1}^{N_c} P(T(X) = \mathbf{x}_i^c, X = \mathbf{x}_k^c | A = a)$$

$$= \sum_{k=1}^{N_c} P(T(X) = \mathbf{x}_i^c | X = \mathbf{x}_k^c, \underbrace{A = a, Y = 1}_{\text{conditional independence}}) P(X = \mathbf{x}_k^c | A = a, Y = 1) = \sum_{k=1}^{N_c} \mathbf{T}[i,k] \mathbf{p}_{a,1}[k].$$

(27)

Plugging (27) into (26), we obtain:

$$P(\hat{Y}^F(T(X)) = 1 | A = a) = \sum_{k=1}^{N_c} \mathbf{s}^F[i] \sum_{k=1}^{N_c} \mathbf{T}[i,k] \mathbf{p}_{a,1}[k] = \sum_{k=1}^{N_c} \sum_{k=1}^{N_c} \mathbf{T}[i,k] \mathbf{s}^F[i] \mathbf{p}_{a,1}[k] = \mathbf{p}_{a,1}^T \mathbf{T} \mathbf{s}^F.$$

Through similar derivation, we can find that $P(\hat{Y}^F(T(X)) = 1 | A = b, Y = 1) = \mathbf{p}_{b,1}^T \mathbf{T} \mathbf{s}^F$. Thus,

$$|P(\hat{Y}^F(T(X)) = 1 | A = a, Y = 1) - P(\hat{Y}^F(T(X)) = 1 | A = b, Y = 1)| \le \epsilon_{EOp}$$

is equivalent to

$$|(\mathbf{p}_{a,1} - \mathbf{p}_{b,1})^T \mathbf{T} \mathbf{s}^F| \le \epsilon_{EOp}.$$

## D.4 PREDICTIVE EQUALITY

Observing that $P(\hat{Y}^F(T(X)) = 0 | T(X) = \mathbf{x}_i^c) = 1 - \mathbf{s}^F[i]$, we simply replace this in the Equal Opportunity derivation to obtain:

$$|(\mathbf{p}_{a,0} - \mathbf{p}_{b,0})^T \mathbf{T} (\mathbf{1}_{N_c} - \mathbf{s}^F)| \le \epsilon_{EOp}.$$

## D.5 EQUAL ACCURACY:

Observe that:

$$P(\hat{Y}^F(T(X)) = Y | A = a) = P(\hat{Y}^F(T(X)) = 1, Y = 1 | A = a) + P(\hat{Y}^F(T(X)) = 0, Y = 0 | A = a)$$

$$= \sum_{i=1}^{N_c} \underbrace{P(\hat{Y}^F(T(X)) = 1, Y = 1, T(X) = \mathbf{x}_i^c | A = a)}_{(A)} + \underbrace{P(\hat{Y}^F(T(X)) = 0, Y = 0, T(X) = \mathbf{x}_i^c | A = a)}_{(B)}$$

(28)

The derivations for $(A)$ and $(B)$ are similar, so we focus on deriving $(A)$ here.

$$P(\hat{Y}^F(T(X)) = 1, Y = 1, T(X) = \mathbf{x}_i^c | A = a) =$$

$$= \overbrace{P(\hat{Y}^F(T(X)) = 1 | T(X) = \mathbf{x}_i^c, \underbrace{A = a, Y = 1}_{\text{conditional independence}})}^{\mathbf{s}^F[i] \text{ by definition.}} P(T(X) = \mathbf{x}_i^c, Y = 1 | A = a) \quad (29)$$

Next, note that

$$P(T(X) = \mathbf{x}_i^c, Y = 1 | A = a) = \sum_{k=1}^{N_c} P(T(X) = \mathbf{x}_i^c, Y = 1, X = \mathbf{x}_k^c | A = a)$$

$$= \sum_{k=1}^{N_c} \underbrace{P(T(X) = \mathbf{x}_i^c | X = \mathbf{x}_k^c, \underbrace{Y = 1, A = a}_{\text{conditional independence}})}_{\mathbf{T}[i,k]} \underbrace{P(Y = 1, X = \mathbf{x}_k^c | A = a)}_{\mathbf{p}_a^1[k]} = \sum_{k=1}^{N_c} \mathbf{T}[i,k] \mathbf{p}_a^1[k].$$

(30)

Plugging (30) into (29), we have that $(A)$ is given by:

$$P(\hat{Y}^F(T(X)) = 1, Y = 1, T(X) = \mathbf{x}_i^c | A = a) = \mathbf{s}^F[i] \sum_{k=1}^{N_c} \mathbf{T}[i, k] \mathbf{p}_a^1[k]. \tag{31}$$

Through a similar derivation, coupled with the fact that $P(\hat{Y}^F(T(X)) = 0 | T(X) = \mathbf{x}_i^c, A = a, Y = 0) = 1 - \mathbf{s}^F[i]$, we can find that $(B)$ is given by:

$$P(\hat{Y}^F(T(X)) = 0, Y = 0, T(X) = \mathbf{x}_i^c | A = a) = (1 - \mathbf{s}^F[i]) \sum_{k=1}^{N_c} \mathbf{T}[i, k] \mathbf{p}_a^0[k]. \tag{32}$$

Finally, substituting (31) and (32) into (28), we obtain that

$$P(\hat{Y}^F(T(X)) = Y | A = a) = \sum_{i=1}^{N_c} \mathbf{s}^F[i] \sum_{k=1}^{N_c} \mathbf{T}[i, k] \mathbf{p}_a^1[k] + (1 - \mathbf{s}^F[i]) \sum_{k=1}^{N_c} \mathbf{T}[i, k] \mathbf{p}_a^0[k]$$

$$= \sum_{i=1}^{N_c} \sum_{k=1}^{N_c} \mathbf{T}[i, k] \mathbf{s}^F[i] \mathbf{p}_a^1[k] + \sum_{i=1}^{N_c} \sum_{k=1}^{N_c} \mathbf{T}[i, k] (1 - \mathbf{s}^F[i]) \mathbf{p}_a^0[k]$$

$$= \mathbf{p}_a^{1T} \mathbf{T} \mathbf{s}^F + \mathbf{p}_a^{0T} \mathbf{T} (\mathbf{1}_{N_c} - \mathbf{s}^F).$$

Through similar derivation, we obtain $P(\hat{Y}^F(T(X)) = Y | A = b) = \mathbf{p}_b^{1T} \mathbf{T} \mathbf{s}^F + \mathbf{p}_b^{0T} \mathbf{T} (\mathbf{1}_{N_c} - \mathbf{s}^F)$. Thus,

$$|P(\hat{Y}^F(T(X)) = Y | A = a) - P(\hat{Y}^F(T(X)) = Y | A = b)| \leq \epsilon_{EA}$$

is equivalent to

$$|(\mathbf{p}_a^1 - \mathbf{p}_b^1)^T \mathbf{T} \mathbf{s}^F + (\mathbf{p}_a^0 - \mathbf{p}_b^0)^T \mathbf{T} (\mathbf{1}_{N_c} - \mathbf{s}^F)| \leq \epsilon_{EA}$$

### D.6 LOCAL INDIVIDUAL FAIRNESS:

Let $i$ and $j$ represent the arbitrary $n^{th}$ pair of cell centroids that satisfy $d_{\mathcal{X}}(\mathbf{x}_i^c, \mathbf{x}_j^c) \leq \eta$. Observe that the probability that a feature vector, $\mathbf{x}_i^c$, is classified as 1 by $S^F$ after it has been transformed by $T$ is given by:

$$t[i] = E[S^F(T(X))|X = \mathbf{x}_i^c] = \sum_{k=1}^{N_c} S^F(\mathbf{x}_k^c) P(T(X) = \mathbf{x}_k^c | X = \mathbf{x}_i^c) = \sum_{k=1}^{N_c} \mathbf{s}^F[k] \mathbf{T}[k, i].$$

Similarly, $t[j] = \sum_{k=1}^{N_c} \mathbf{s}^F[k] \mathbf{T}[k, j]$. Now, let $\mathbf{w}_n$ be a vector in which every entry is zero, except for the $i^{th}$ and $j^{th}$ entries, which contain values of $e^{-\theta d_{\mathcal{X}}^2(\mathbf{x}_i, \mathbf{x}_j)}$ and $-e^{-\theta d_{\mathcal{X}}^2(\mathbf{x}_i, \mathbf{x}_j)}$, respectively. Then,

$$e^{-\theta d_{\mathcal{X}}^2(\mathbf{x}_i, \mathbf{x}_j)} \left| t^F[i] - t^F[j] \right| = e^{-\theta d_{\mathcal{X}}^2(\mathbf{x}_i, \mathbf{x}_j)} \left| \sum_{k=1}^{N_c} \mathbf{s}^F[k] (\mathbf{T}[k, i] - \mathbf{T}[k, j]) \right|$$

$$\left| \sum_{k=1}^{N_c} \left( \mathbf{s}^F[k] (\mathbf{T}[k, i] e^{-\theta d_{\mathcal{X}}^2(\mathbf{x}_i, \mathbf{x}_j)} - \mathbf{T}[k, j] e^{-\theta d_{\mathcal{X}}^2(\mathbf{x}_i, \mathbf{x}_j)}) \right) \right| = \left| \sum_{l=1}^{N_c} \sum_{k=1}^{N_c} \mathbf{s}^F[k] \mathbf{T}[k, i] \mathbf{w}_n[l] \right|$$

$$= \left| \mathbf{w}_n^T \mathbf{T} \mathbf{s}^F \right|.$$

Thus,

$$e^{-\theta d_{\mathcal{X}}(\mathbf{x}_i, \mathbf{x}_j)} |t^F[i] - t^F[j]| \leq \epsilon_{IF}$$

is equivalent to:

$$\left| \mathbf{w}_n^T \mathbf{T} \mathbf{s}^F \right| \leq \epsilon_{IF}.$$

To enforce this constraint for all $i$ and $j$, such that $d_{\mathcal{X}}(\mathbf{x}_i, \mathbf{x}_j) \leq \eta$, simply stack all such $\mathbf{w}_n^T$ into a matrix $\mathbf{W}$. The associated constraint is then given by:

$$\left| \mathbf{W} \mathbf{T} \mathbf{s}^F \right| \leq \mathbf{1}_{N_c} \epsilon_{IF}.$$

# E    PROOF OF CLAIM 1

We will now show that the following minimization problem is convex:

$$\max_{\boldsymbol{\rho}} \min_{\mathbf{T} \in \mathcal{P}^{N_c \times N_c}} -\lambda(\mathbf{s}^{FT}\mathbf{T}\mathbf{p}^1 + (\mathbf{1}_{N_c} - \mathbf{s}^F)^T\mathbf{T}\mathbf{p}^0) + \beta\|\mathbf{T}(\mathbf{p}_a - \mathbf{p}_b)\|_1$$

$$+ \langle \boldsymbol{\rho}, \max(\tilde{f}(\mathbf{T}) - \tilde{\mathbf{f}}, \mathbf{0}_{2(N_{Nbr}+4)})\rangle + \frac{\tau}{2}\|\max(\tilde{f}(\mathbf{T}) - \tilde{\mathbf{f}}, \mathbf{0}_{2(N_{Nbr}+4)})\|_2^2.$$

*Proof.* Observing that the sum of convex functions is convex, it suffices to prove that this minimization problem is convex by showing that every term in the problem is convex in $\mathbf{T}$, which we will now do. $-\lambda(\mathbf{s}^{FT}\mathbf{T}\mathbf{p}^1 + (\mathbf{1}_{N_c} - \mathbf{s}^F)^T\mathbf{T}\mathbf{p}^0)$ is linear in $\mathbf{T}$, and thus convex. $\beta\|\mathbf{T}(\mathbf{p}_a - \mathbf{p}_b)\|_1$ is convex since the composition of a convex function with an affine function is convex.

To show that the final terms in the minimization problem are convex, we begin by showing that $g(\mathbf{T}) = \max(\tilde{f}(\mathbf{T}) - \tilde{\mathbf{f}}, \mathbf{0}_{2(N_{Nbr}+4)})$ is elementwise convex using the definition of convexity. Specifically, for $\theta \in [0, 1]$ and $\mathbf{T}_1, \mathbf{T}_2 \in \mathcal{P}^{N_c \times N_c}$, observe that:

$$g(\theta\mathbf{T}_1 + (1-\theta)\mathbf{T}_2) = \max(\tilde{f}(\theta\mathbf{T}_1 + (1-\theta)\mathbf{T}_2) - \tilde{\mathbf{f}}, \ \mathbf{0}_{2(N_{Nbr}+4)})$$

$$= \max(\underbrace{\tilde{\mathbf{P}}}_{\begin{bmatrix}-\mathbf{P}\\\mathbf{P}\end{bmatrix}} \left[\mathbf{M} \circ \left(\tilde{\mathbf{I}}(\theta\mathbf{T}_1 + (1-\theta)\mathbf{T}_2)\tilde{\mathbf{I}}^T\right)\right]\mathbf{s}^F - \tilde{\mathbf{f}}, \ \mathbf{0}_{2(N_{Nbr}+4)})$$

$$= \max(\theta\underbrace{\left\{\tilde{\mathbf{P}}\left[\mathbf{M} \circ \left(\tilde{\mathbf{I}}\mathbf{T}_1\tilde{\mathbf{I}}^T\right)\right]\mathbf{s}^F - \tilde{\mathbf{f}}\right\}}_{\tilde{f}(\mathbf{T}_1)} + (1-\theta)\underbrace{\left\{\tilde{\mathbf{P}}\left[\mathbf{M} \circ \left(\tilde{\mathbf{I}}\mathbf{T}_2\tilde{\mathbf{I}}^T\right)\right]\mathbf{s}^F - \tilde{\mathbf{f}}\right\}}_{\tilde{f}(\mathbf{T}_2)}, \ \mathbf{0}_{2(N_{Nbr}+4)})$$

$$= \max(\theta\tilde{f}(\mathbf{T}_1) + (1-\theta)\tilde{f}(\mathbf{T}_2), \ \mathbf{0}_{2(N_{Nbr}+4)})$$

$$\underbrace{\leq}_{\substack{\text{Elementwise by convexity}\\\text{of pointwise max.}}} \max(\theta\tilde{f}(\mathbf{T}_1), \ \mathbf{0}_{2(N_{Nbr}+4)}) + \max((1-\theta)\tilde{f}(\mathbf{T}_2), \ \mathbf{0}_{2(N_{Nbr}+4)})$$

$$= \theta\max(\tilde{f}(\mathbf{T}_1), \ \mathbf{0}_{2(N_{Nbr}+4)}) + (1-\theta)\max(\tilde{f}(\mathbf{T}_2), \ \mathbf{0}_{2(N_{Nbr}+4)})$$

$$= \theta g(\mathbf{T}_1) + (1-\theta)g(\mathbf{T}_2)$$

Thus, the inner product between $\boldsymbol{\rho}$ and $g(\mathbf{T})$ is simply a linear combination of convex functions, making the second to last term in the optimization problem convex. Finally, the convexity of the last term in the optimization problem follows from the fact that $g(\mathbf{T})$ is a non-negative convex function and the norm is convex and non-decreasing over the set $\mathbb{R}_{\geq 0}$. $\qquad\square$

# F    FAIRNESS-ACCURACY TRADEOFF (SENSITIVE AWARE)

In Section 3.1, the minimization problem for analyzing the tradeoff between accuracy and fairness was formulated under the assumption that the sensitive attribute is unavailable. In this section, we adapt this formulation for the situation in which the sensitive attribute is allowed to be used by the unconstrained and fair Bayesian classifiers to make its decisions. With this information, the solution of the Bayesian classifier is given by $\mathbf{s}^B = \begin{bmatrix}\mathbf{s}_a^{BT}, & \mathbf{s}_b^{BT}\end{bmatrix}^T$, where

$$\mathbf{s}_a^B[i] = \underset{y}{\arg\max}\, \mathbf{p}_a^y[i] \qquad \text{and} \qquad \mathbf{s}_b^B[i] = \underset{y}{\arg\max}\, \mathbf{p}_b^y[i], \forall i,$$

which again produces binary decisions. The accuracy of this classifier is given by $Acc_b = \sum_{g \in \mathcal{A}}\sum_{i=1}^{N_c}\mathbf{p}^{\mathbf{s}_g^B[i]}[i]$. Constructing the minimization problem for finding the optimal fair Bayesian oracles's decisions is extended from the minimization problem presented in Section 3.1 by separating its decisions by group. Towards this end, we let $\mathbf{m}_g = \mathbf{s}_g^B - \mathbf{s}_g^F, g \in \mathcal{A}$, and $\mathcal{I}_g^+$ and $\mathcal{I}_g^-$, be the set of indices associated with the positive and negative elements of

$\mathbf{p}^{1,g} - \mathbf{p}^{0,g}$, respectively. Then, the minimization problem is given by:

$$\min_{\{\mathbf{m}_g\}} \sum_{g \in \mathcal{A}} (\mathbf{p}^{1,g} - \mathbf{p}^{0,g})^T \mathbf{m}_g,$$

$$|\mathbf{p}_a^T(\mathbf{s}_a^B - \mathbf{m}_a) - \mathbf{p}_b^T(\mathbf{s}_b^B - \mathbf{m}_b)| \leq \epsilon_{DP} \qquad (DP)$$

$$|\mathbf{p}_{a,0}^T(\mathbf{s}_a^B - \mathbf{m}_a) - \mathbf{p}_{b,0}^T(\mathbf{s}_b^B - \mathbf{m}_b)| \leq \epsilon_{PE} \qquad (PE)$$

$$|\mathbf{p}_{a,1}^T(\mathbf{s}_a^B - \mathbf{m}_a) - \mathbf{p}_{b,1}^T(\mathbf{s}_b^B - \mathbf{m}_b)| \leq \epsilon_{EOp} \qquad (EOp)$$

$$\epsilon_{EOp} = \epsilon_{PE} \qquad (EOd)$$

$$|\mathbf{p}_a^{1T}(\mathbf{s}_a^B - \mathbf{m}_a) - \mathbf{p}_b^{1T}(\mathbf{s}_b^B - \mathbf{m}_b) +$$
$$\mathbf{p}_a^{0T}(\mathbf{1}_{N_c} - \mathbf{s}_a^B + \mathbf{m}_a) - \mathbf{p}_b^{0T}(\mathbf{1}_{N_c} - \mathbf{s}_b^B + \mathbf{m}_b)| \leq \epsilon_{EA} \qquad (EA)$$

$$|\mathbf{W} \sum_{g \in \mathcal{A}} \mathrm{diag}(\mathbf{p}_g)(\mathbf{s}_g^B - \mathbf{m}_g)| \leq \epsilon_{IF} \mathbf{1}_{N_c}, \qquad (Ind)$$

$$0 \leq \mathbf{m}_g[i] \leq 1, \quad i \in \mathcal{I}_g^+, \quad g \in \mathcal{A}$$

$$-1 \leq \mathbf{m}_g[i] \leq 0, \quad i \in \mathcal{I}_g^-, \quad g \in \mathcal{A}, \qquad (33)$$

where $\mathbf{W}$ is defined as in equation (4). The derivation of each of these terms is similar to the derivation provided in Appendix C, with the added step of expanding each term using group affiliation information.

## G    TRANSFER FAIRNESS TO DECORRELATED DOMAIN (SENSITIVE AWARE)

In Section 3.2 the minimization problem for transferring fairness to a decorrelated domain was formulated under the assumption of unawareness of the sensitive attribute. Availability of the sensitive attribute provides more flexibility in construction of the mapping, allowing us to transform the feature vectors belonging to each group using group-specific transformations. Specifically, given a fixed fair Bayesian Classifier's decision map, $\mathbf{s}^F = \begin{bmatrix} \mathbf{s}_a^{FT}, & \mathbf{s}_b^{FT} \end{bmatrix}^T$, where the first $N_c$ entries represent the scores associated with Group $a$ and the remaining $N_c$ entries represent the scores for Group $b$, we aim to optimize for two mixing matrices, $\mathbf{T}_a, \mathbf{T}_b \in \mathcal{P}^{2N_c \times N_c}$, using the following optimization problem:

$$\min_{\mathbf{T}_a, \mathbf{T}_b \in \mathcal{P}^{2N_c \times N_c}} - \lambda \sum_{g \in \mathcal{A}} (\mathbf{s}^{FT} \mathbf{T}_g \mathbf{p}^{1,g} + (\mathbf{1}_{N_c} - \mathbf{s}^F)^T \mathbf{T}_g \mathbf{p}^{0,g})$$
$$+ \beta \|\mathbf{T}_a \mathbf{p}_a - \mathbf{T_b} \mathbf{p}_b\|_1$$
$$\text{s.t.} \quad |f(\mathbf{T}_a, \mathbf{T}_b)| \leq \mathbf{F} \quad \textit{(Fairness condition)}. \qquad (34)$$

The derivation of each of these terms is similar to the derivation provided in Appendix D, with the added step of expanding each term using group affiliation information. The terms in the objective function and the $(Fairness)$ constraint for this optimization problem still have the same interpretation as in the situation of unawareness of the sensitive attribute. Note that in this case, the row dimension of the mixing matrices, $\mathbf{T}_a$ and $\mathbf{T}_b$, are twice that of the mixing matrix, $\mathbf{T}$, from minimization problem (6). This is because, each VQ cell effectively has two bins of information associated with it (one for each group in the original space of feature vectors). Nevertheless, optimization over these mixing matrices ensures that in the decorrelated domain, all bins of all VQ cells are conditionally independent of the sensitive attribute. The function $f(\mathbf{T}_a, \mathbf{T}_b) \in \mathbb{R}^{1 \times (N_{Nbr}+4)}$ is now given by the following equation:

$$f(\mathbf{T}_a, \mathbf{T}_b) = \underbrace{\begin{bmatrix} \mathbf{s}^F & (\mathbf{1}_{N_c} - \mathbf{s}^F) \end{bmatrix}}_{\tilde{\mathbf{s}}^{FT}} \left( \underbrace{\begin{bmatrix} \mathbf{T}_a & \mathbf{O}_{2N_c, N_c} \\ \mathbf{O}_{2N_c, N_c} & \mathbf{T}_a \end{bmatrix}}_{\tilde{\mathbf{T}}_a} \underbrace{\begin{bmatrix} \mathbf{p}_a & \mathbf{p}_{a1} & \mathbf{0}_{N_c} & \mathbf{p}_a^1 & \mathbf{W}_a^T \\ \mathbf{0}_{N_c} & \mathbf{0}_{N_c} & \mathbf{p}_{a0} & \mathbf{p}_a^0 & \mathbf{O}_{N_{Nbr}, N_c} \end{bmatrix}}_{\mathbf{P}_a} \right.$$

$$\left. - \underbrace{\begin{bmatrix} \mathbf{T}_b & \mathbf{O}_{2N_c, N_c} \\ \mathbf{O}_{2N_c, N_c} & \mathbf{T}_b \end{bmatrix}}_{\tilde{\mathbf{T}}_b} \underbrace{\begin{bmatrix} \mathbf{p}_b & \mathbf{p}_{b1} & \mathbf{0}_{N_c} & \mathbf{p}_b^1 & -\mathbf{W}_b^T \\ \mathbf{0}_{N_c} & \mathbf{0}_{N_c} & \mathbf{p}_{b0} & \mathbf{p}_b^0 & \mathbf{O}_{N_{Nbr}, N_c} \end{bmatrix}}_{\mathbf{P}_b} \right)$$

$$= \tilde{\mathbf{s}}^{FT}(\tilde{\mathbf{T}}_a \mathbf{P}_a - \tilde{\mathbf{T}}_b \mathbf{P}_b)$$

where, $\tilde{\mathbf{T}}_g$ can be directly written as a function of $\mathbf{T}_g$:

$$\tilde{\mathbf{T}}_g = \underbrace{\begin{bmatrix} \mathbf{1}_{2N_c \times N_c} & \mathbf{O}_{2N_c \times N_c} \\ \mathbf{O}_{2N_c \times N_c} & \mathbf{1}_{2N_c \times N_c} \end{bmatrix}}_{\tilde{\mathbf{M}}} \circ \left( \underbrace{\begin{bmatrix} \mathbf{I}_{2N_c} \\ \mathbf{I}_{2N_c} \end{bmatrix}}_{\tilde{\mathbf{I}}_2} \mathbf{T}_g \underbrace{[\mathbf{I}_{N_c} \quad \mathbf{I}_{N_c}]}_{\tilde{\mathbf{I}}_1^T} \right).$$

Still, each of the first four elements of $|f(\mathbf{T}_a, \mathbf{T}_b)|$ captures the degree to which a particular group fairness notion is violated in the transformed space, while the remaining $N_{nbr}$ terms enforce individual fairness. Setting $\mathbf{F} = [\epsilon_{DP}, \ \epsilon_{PE}, \ \epsilon_{EOp}, \ \epsilon_{EOd}, \ \mathbf{1}_{N_{nbr}}^T \epsilon_{IF}]$, preserves the group and individual fairness constraints.

Again, we can reformulate the $(Fairness)$ constraint as an equality constraint of the form: $\max(\tilde{f}(\mathbf{T}_a, \mathbf{T}_b) - \tilde{\mathbf{F}}, \mathbf{O}_{1 \times 2(N_{Nbr}+4)}) = \mathbf{O}_{1 \times 2(N_{Nbr}+4)}$, where

$$\tilde{f}(\mathbf{T}_a, \mathbf{T}_b) = \tilde{\mathbf{s}}^{FT}(\tilde{\mathbf{T}}_\mathbf{a} \underbrace{[\mathbf{P}_a \quad -\mathbf{P}_a]}_{\tilde{\mathbf{P}}_a} - \tilde{\mathbf{T}}_\mathbf{b} \underbrace{[\mathbf{P}_b \quad -\mathbf{P}_b]}_{\tilde{\mathbf{P}}_b}) \qquad \text{and} \qquad \tilde{\mathbf{F}} = [\mathbf{F} \quad \mathbf{F}]$$

The final Augmented Lagrangian can be formed as:

$$\max_{\boldsymbol{\rho}} \min_{\mathbf{T}_a, \mathbf{T}_b \in \mathcal{P}^{2N_c \times N_c}} \underbrace{-\lambda \sum_{g \in \mathcal{A}} (\mathbf{s}^{FT} \mathbf{T}_g \mathbf{p}^{1,g} + (\mathbf{1}_{N_c} - \mathbf{s}^F)^T \mathbf{T}_g \mathbf{p}^{0,g})}_{Acc_d(\mathbf{T}_a, \mathbf{T}_b)} + \beta \underbrace{\|\mathbf{T}_a \mathbf{p}_a - \mathbf{T}_\mathbf{b} \mathbf{p}_b\|_1}_{L_d(\mathbf{T}_a, \mathbf{T}_b)}$$

$$+ \underbrace{\langle \boldsymbol{\rho}, \max(\tilde{f}(\mathbf{T}_a, \mathbf{T}_b) - \tilde{\mathbf{F}}, \mathbf{O}_{1 \times 2(N_{Nbr}+4)}) \rangle}_{Aug_1(\mathbf{T}_a, \mathbf{T}_b)}$$

$$+ \frac{\tau}{2} \underbrace{\| \max(\tilde{f}(\mathbf{T}_a, \mathbf{T}_b) - \tilde{\mathbf{F}}, \mathbf{O}_{1 \times 2(N_{Nbr}+4)}) \|_F^2}_{Aug_2(\mathbf{T}_a, \mathbf{T}_b)} \qquad (35)$$

Minimization problem (35) is convex according to Claim 2. Solving this minimization problem is equivalent to solving minimization problem (34) and can be done by applying the alternating direction method of multipliers (ADMM) to optimize for $\mathbf{T}_a$ and $\mathbf{T}_b$ until convergence using the following updates.

$$\mathbf{T}_a^{k+1} = \underset{\mathbf{T}_a \in \mathcal{P}^{N_c \times N_c}}{\operatorname{argmin}} L(\mathbf{T}_a, \mathbf{T}_b^k, \boldsymbol{\rho}^k)$$

$$\mathbf{T}_b^{k+1} = \underset{\mathbf{T}_b \in \mathcal{P}^{N_c \times N_c}}{\operatorname{argmin}} L(\mathbf{T}_a^{k+1}, \mathbf{T}_b, \boldsymbol{\rho}^k)$$

$$\boldsymbol{\rho}^{k+1} = \boldsymbol{\rho}^k + \tau \max(\tilde{f}(\mathbf{T}_a^{k+1}, \mathbf{T}_b^{k+1}) - \tilde{\mathbf{F}}, \mathbf{O}_{1 \times 2(N_{Nbr}+4)}).$$

**Claim 2.** *Minimization problem (35) is jointly convex in* $\mathbf{T}_a$ *and* $\mathbf{T}_b$

*Proof.* Observing that the sum of jointly convex functions is jointly convex, it suffices to prove that this minimization problem is jointly convex if every term in the problem is jointly convex in $\mathbf{T}_a$ and $\mathbf{T}_b$. Thus, we will show that each term in the problem is jointly convex in $\mathbf{T}_a$ and $\mathbf{T}_b$. Consider $\mathbf{T}_{1,a}, \mathbf{T}_{2,a}, \mathbf{T}_{1,b}, \mathbf{T}_{2,b} \in \mathcal{P}^{2N_c \times N_c}$ and $\theta \in [0,1]$.

Joint convexity of $Acc_d(\mathbf{T}_a, \mathbf{T}_b)$:

$$Acc_d(\theta \mathbf{T}_{1,a} + (1-\theta)\mathbf{T}_{2,a}, \theta \mathbf{T}_{1,b} + (1-\theta)\mathbf{T}_{2,b})$$

$$= \sum_{g \in \mathcal{A}} (\mathbf{s}^{FT}(\theta \mathbf{T}_{1,g} + (1-\theta)\mathbf{T}_{2,g})\mathbf{p}^{1,g} + (\mathbf{1}_{N_c} - \mathbf{s}^F)^T(\theta \mathbf{T}_{1,g} + (1-\theta)\mathbf{T}_{2,g})\mathbf{p}^{0,g})$$

$$= \theta \sum_{g \in \mathcal{A}} \{\mathbf{s}^{FT} \mathbf{T}_{1,g} \mathbf{p}^{1,g} + (\mathbf{1}_{N_c} - \mathbf{s}^F)^T \mathbf{T}_{1,g} \mathbf{p}^{0,g}\}$$

$$+ (1-\theta) \sum_{g \in \mathcal{A}} \{\mathbf{s}^{FT} \mathbf{T}_{2,g} \mathbf{p}^{2,g} + (\mathbf{1}_{N_c} - \mathbf{s}^F)^T \mathbf{T}_{2,g} \mathbf{p}^{0,g}\}$$

$$= \theta Acc_d(\mathbf{T}_{1,a}, \mathbf{T}_{1,b}) + (1-\theta)Acc_d(\mathbf{T}_{2,a}, \mathbf{T}_{2,b})$$

Joint convexity of $L_d(\mathbf{T}_a, \mathbf{T}_b)$:

$$L_d(\theta\mathbf{T}_{1,a} + (1-\theta)\mathbf{T}_{2,a}, \theta\mathbf{T}_{1,b} + (1-\theta)\mathbf{T}_{2,b})$$
$$= \|(\theta\mathbf{T}_{1,a} + (1-\theta)\mathbf{T}_{2,a})\mathbf{p}_a - (\theta\mathbf{T}_{1,b} + (1-\theta)\mathbf{T}_{2,b})\mathbf{p}_b\|_1$$
$$= \|\theta(\mathbf{T}_{1,a}\mathbf{p}_a - \mathbf{T}_{1,b}\mathbf{p}_b) + (1-\theta)(\mathbf{T}_{2,a}\mathbf{p}_a - \mathbf{T}_{2,b}\mathbf{p}_b)\|_1$$
$$\underset{\substack{\text{Triangle}\\\text{Inequality}}}{\leq} \|\theta(\mathbf{T}_{1,a}\mathbf{p}_a - \mathbf{T}_{1,b}\mathbf{p}_b)\|_1 + \|(1-\theta)(\mathbf{T}_{2,a}\mathbf{p}_a - \mathbf{T}_{2,b}\mathbf{p}_b)\|_1$$
$$= \theta\|\mathbf{T}_{1,a}\mathbf{p}_a - \mathbf{T}_{1,b}\mathbf{p}_b\|_1 + (1-\theta)\|\mathbf{T}_{2,a}\mathbf{p}_a - \mathbf{T}_{2,b}\mathbf{p}_b\|_1$$
$$= \theta L_d(\mathbf{T}_{1,a}, \mathbf{T}_{1,b}) + (1-\theta)L_d(\mathbf{T}_{2,a}, \mathbf{T}_{2,b})$$

Joint convexity of $Aug_1(\mathbf{T}_a, \mathbf{T}_b)$:

We will show that every element of $\tilde{g}(\mathbf{T}_a, \mathbf{T}_b) = \max(\tilde{f}(\mathbf{T}_a, \mathbf{T}_b) - \tilde{\mathbf{F}}, \mathbf{O}_{1\times 2(N_{Nbr}+4)})$ is jointly convex in $\mathbf{T}_a$ and $\mathbf{T}_b$. It directly follows from this that $Aug_1(\mathbf{T}_a, \mathbf{T}_b)$ is jointly convex in $\mathbf{T}_a$ and $\mathbf{T}_b$ since the inner product is a linear combination of jointly convex functions.

$$\tilde{g}(\theta\mathbf{T}_{1,a} + (1-\theta)\mathbf{T}_{2,a}, \theta\mathbf{T}_{1,b} + (1-\theta)\mathbf{T}_{2,b})$$
$$= \max(\tilde{f}(\theta\mathbf{T}_{1,a} + (1-\theta)\mathbf{T}_{2,a}, \theta\mathbf{T}_{1,b} + (1-\theta)\mathbf{T}_{2,b}) - \tilde{\mathbf{F}}, \mathbf{O}_{1\times 2(N_{Nbr}+4)})$$
$$= \max(\hat{\mathbf{s}}^F\big[\tilde{\mathbf{M}} \circ (\tilde{\mathbf{I}}_2(\theta\mathbf{T}_{1,a} + (1-\theta)\mathbf{T}_{2,a})\tilde{\mathbf{I}}_1^T)\big]\tilde{\mathbf{P}}_a$$
$$\qquad - \hat{\mathbf{s}}^F\big[\tilde{\mathbf{M}} \circ (\tilde{\mathbf{I}}_2(\theta\mathbf{T}_{1,b} + (1-\theta)\mathbf{T}_{2,b})\tilde{\mathbf{I}}_1^T)\big]\tilde{\mathbf{P}}_b - \tilde{\mathbf{F}}, \mathbf{O}_{1\times 2(N_{Nbr}+4)})$$
$$= \max(\theta\{\hat{\mathbf{s}}^F\big[\tilde{\mathbf{M}} \circ (\tilde{\mathbf{I}}_2\mathbf{T}_{1,a}\tilde{\mathbf{I}}_1^T)\big]\tilde{\mathbf{P}}_a - \tilde{\mathbf{M}} \circ (\tilde{\mathbf{I}}_2\theta\mathbf{T}_{1,b}\tilde{\mathbf{I}}_1^T)\big]\tilde{\mathbf{P}}_b - \tilde{\mathbf{F}}\}$$
$$\qquad + (1-\theta)\{\hat{\mathbf{s}}^F\big[\tilde{\mathbf{M}} \circ (\tilde{\mathbf{I}}_2\mathbf{T}_{2,a}\tilde{\mathbf{I}}_1^T)\big]\tilde{\mathbf{P}}_a - \tilde{\mathbf{M}} \circ (\tilde{\mathbf{I}}_2\theta\mathbf{T}_{2,b}\tilde{\mathbf{I}}_1^T)\big]\tilde{\mathbf{P}}_b - \tilde{\mathbf{F}}\}, \mathbf{O}_{1\times 2(N_{Nbr}+4)})$$
$$\underset{\text{Elementwise}}{\leq} \theta\max(\hat{\mathbf{s}}^F\big[\tilde{\mathbf{M}} \circ (\tilde{\mathbf{I}}_2\mathbf{T}_{1,a}\tilde{\mathbf{I}}_1^T)\big]\tilde{\mathbf{P}}_a - \tilde{\mathbf{M}} \circ (\tilde{\mathbf{I}}_2\theta\mathbf{T}_{1,b}\tilde{\mathbf{I}}_1^T)\big]\tilde{\mathbf{P}}_b - \tilde{\mathbf{F}}, \mathbf{O}_{1\times 2(N_{Nbr}+4)})$$
$$\qquad + (1-\theta)\max(\hat{\mathbf{s}}^F\big[\tilde{\mathbf{M}} \circ (\tilde{\mathbf{I}}_2\mathbf{T}_{2,a}\tilde{\mathbf{I}}_1^T)\big]\tilde{\mathbf{P}}_a - \tilde{\mathbf{M}} \circ (\tilde{\mathbf{I}}_2\theta\mathbf{T}_{2,b}\tilde{\mathbf{I}}_1^T)\big]\tilde{\mathbf{P}}_b - \tilde{\mathbf{F}}, \mathbf{O}_{1\times 2(N_{Nbr}+4)})$$
$$= \theta\max(\tilde{f}(\mathbf{T}_{1,a}, \mathbf{T}_{1,b}), \mathbf{O}_{1\times 2(N_{Nbr}+4)}) + (1-\theta)\max(\tilde{f}(\mathbf{T}_{2,a}, \mathbf{T}_{2,b}), \mathbf{O}_{1\times 2(N_{Nbr}+4)})$$
$$= \theta\tilde{g}(\mathbf{T}_{1,a}, \mathbf{T}_{1,b}) + (1-\theta)\tilde{g}(\mathbf{T}_{2,a}, \mathbf{T}_{2,b})$$

Joint convexity of $Aug_2(\mathbf{T}_a, \mathbf{T}_b)$:

The convexity of this term follows from the fact that $\tilde{g}(\mathbf{T}_a, \mathbf{T}_b)$ is a non-negative jointly convex function in $\mathbf{T}_a$ and $\mathbf{T}_b$ and the norm is convex and non-decreasing over the set $\mathbb{R}_{\geq 0}$. $\qquad\square$

## H    EXPERIMENTAL DETAILS

Our experiments are conducted on three benchmark tabular datasets with known biases in the sensitive attributes; namely, the (1) Adult (Kohavi et al. (1996)), (2) Law (Wightman (1998)), and (3) Dutch Census (Van der Laan (2001)) datasets. We treat race as the sensitive attribute for the Law dataset and sex as the sensitive attribute in the Adult and Dutch Census datasets. All codes were written in Python using version 3.8. In the remainder of this section, we provide the experimental details used in each of the three core modules of our analysis.

### H.1    GENERATOR AND VECTOR QUANTIZATION IMPLEMENTATION

To model the population distributions of each dataset, we use Conditional Tabular Generative Adversarial Network (CT-GAN) (Xu et al. (2019)), which is the state-of-the-art for generating mixed continuous and discrete tabular data. For each dataset we train a generator for 400 epochs using the publically available code provided by Xu et al. (2019) [1]. Each generator is

---

[1]https://github.com/sdv-dev/CTGAN

then used to produce one million samples for each dataset, on which vector quantization is applied using the Linde, Buzo, Gray (LBG) algorithm, which is a multi-dimensional generalization of the Lloyd-Max algorithm for vector quantization. To produce a cell decomposition of size $N_c$, this iterative algorithm works by specifying an initial set of $N_c$ centroids. In each iteration, a centroid, $\mathbf{x}_j^c$, is updated by taking the average of all samples that are closer to $\mathbf{x}_j^c$ than any other centroid. This iterative process continues until convergence, specified by a relative error tolerance. We use a publicly available implementation of this algorithm [2], specifying a relative error of 0.01 as the stopping criterion for optimizing the VQ cell partitioning. We set $N_c = 256$ for analyzing the population distribution of each dataset in the body of the main paper.

## H.2 SOLVING MINIMIZATION PROBLEM (5) AND (33)

The linear programs used to perform the fairness-accuracy tradeoff analysis in minimization problems (5) and (33) were implemented using the linprog tool from the Scipy Optimize library. For each of the datasets, we normalize the feature vectors for performing the distance calculations associated with local individual fairness. Since each of the analyzed datasets are composed of a mixture of discete and continuous features, we use the Hamming distance to calculate the elementwise distances between discrete entries of a feature vector and the absolute distance to measure the distances between continuous features. All continuous features are normalized to have zero mean and $\frac{1}{2}$ variance so that the maximum distance of each entry is approximately 1 (Wilson & Martinez (1997)). This was done to ensure that each element of a feature vector has approximately equal contribution to the final distance. The average of these distances is taken as the final distance. For all experiments involving local individual fairness constraints, the following procedure was used for choosing $\eta$. The distances between each pair of cell centroids in our discretized population distribution were calculated. Of this set of distances, the $n^{\text{th}}$–percentile was calculated. All pairs of distances smaller than this percentile represent local neighboring cell centroids. Empirically, we set $n = 3.5$ percentiles to not include too few or too many neighbors in a cell's local neighborhood. Furthermore, we set the parameter $\theta = 1$ in the exponential of the ($Ind$) constraints for both minimization problems.

## H.3 SOLVING MINIMIZATION PROBLEM (10) AND (35)

The implementation of the method of multipliers used for solving minimization problem (10) was performed using Tensorflow, version 2.8. To perform the updates specified by equation (11) to $\mathbf{T}$ in each iteration, we perform gradient descent with a momentum of 0.9 and a decaying learning rate from 1e-2 to 1e-12. The entire minimization process is terminated once the sum of square residuals for $\boldsymbol{\rho}$ and $\mathbf{T}$ falls below 1e-4.

A similar process is used to solve minimization problem (35) with the following modifications. In each iteration we sequentially minimize $\mathbf{T}_a$ and $\mathbf{T}_b$ (using the same parameters specified above). Furthermore, the entire minimization process is halted once the sum of square residuals for $\boldsymbol{\rho}$, $\mathbf{T}_a$, and $\mathbf{T}_b$ falls below 1e-4.

## I EXPERIMENTAL RESULTS FOR DECORRELATION ON LAW AND DUTCH DATASETS

In this section, we report the results from our decorrelation analysis under different combinations of fairness notions for the Dutch Census and Law datasets. Again, for all combinations involving individual fairness, we hold $\epsilon_{Ind} = 0.05$, while all group fairness constraints are tested for relaxations of $0.0, 0.5$, and $0.10$. Thus, we report the average over three values in each of the cells in Tables 2 and 3, along with the associated standard deviations in parentheses. Similar to the Adult dataset, the accuracy of the fair Bayesian oracle on the decorrelated feature vectors typically falls by around 2% or less, except in the case of the EA+Ind combination under awareness of the sensitive attribute for both datasets. On

---

[2]https://github.com/internaut/py-lbg

the other hand, prior to decorrelating the space of feature vectors, the value of $L_d$ under awareness of the sensitive attribute was 2 for both datasets, while under unawareness of the sensitive attribute, the respective values of $L_d$ for the Dutch Census and Law datasets were 0.56 and 0.98. Thus, the reported values of $L_d$ for the different fairness combinations for both datasets clearly show substantial improvements in decorrelating the space of feature vectors with respect to the sensitive attribute. As with the Adult dataset, the EA+EOd pairing is the only one that struggles to achieve an $L_d$ value less than 0.1 for both datasets, though even for this pairing, the transformed feature vectors are much more decorrelated with respect to the sensitive attribute compared to the non-transformed feature vectors.

Table 2: Results for transferring fairness to decorrelated domain for for Law dataset.

| | Awareness | | | Unawareness | | |
|---|---|---|---|---|---|---|
| Fairness Measure | Acc. Reduction | | $L_d$ | | Acc. Reduction | | $L_d$ | |
| DP+EA | 0.021 | (0.018) | 0.000 | (0.000) | 0.026 | (0.025) | 0.000 | (0.000) |
| DP+EOd | 0.020 | (0.018) | 0.000 | (0.000) | 0.014 | (0.013) | 0.000 | (0.000) |
| EA+EOd | 0.019 | (0.026) | 0.120 | (0.113) | 0.010 | (0.012) | 0.070 | (0.074) |
| DP+Ind. | 0.021 | (0.013) | 0.000 | (0.000) | 0.005 | (0.005) | 0.000 | (0.000) |
| EA+Ind. | 0.056 | (0.006) | 0.000 | (0.000) | 0.004 | (0.001) | 0.000 | (0.000) |
| EOd+Ind. | 0.020 | (0.006) | 0.023 | (0.021) | 0.005 | (0.004) | 0.000 | (0.000) |

Table 3: Results for transferring fairness to decorrelated domain for Dutch dataset.

| | Awareness | | | Unawareness | | |
|---|---|---|---|---|---|---|
| Fairness Measure | Acc. Reduction | | $L_d$ | | Acc. Reduction | | $L_d$ | |
| DP+EA | 0.017 | (0.016) | 0.000 | (0.000) | 0.025 | (0.023) | 0.016 (0.19) | |
| DP+EOd | 0.009 | (0.016) | 0.000 | (0.000) | 0.015 | (0.022) | 0.062 (0.053) | |
| EA+EOd | 0.021 | (0.019) | 0.199 | (0.099) | 0.019 | (0.017) | 0.193 (0.088) | |
| DP+Ind. | 0.020 | (0.008) | 0.000 | (0.000) | 0.009 | (0.007) | 0.000 (0.000) | |
| EA+Ind. | 0.061 | (0.011) | 0.011 | (0.032) | 0.012 | (0.002) | 0.000 (0.000) | |
| EOd+Ind. | 0.022 | (0.009) | 0.020 | (0.035) | 0.011 | (0.006) | 0.018 (0.031) | |

## J    VERIFYING THE FIDELITY OF GENERATOR

Table 4: PCC and TV Distances between distributions constructed from real and generated data for three datasets. All p-values are below 0.001.

| | PCC | | | TV Distance | | |
|---|---|---|---|---|---|---|
| Distribution | Adult | Law | Dutch | Adult | Law | Dutch |
| $p_{X,A=a,Y=0}(x)$ | 0.96 | 0.98 | 0.94 | 0.04 | 0.01 | 0.05 |
| $p_{X,A=a,Y=1}(x)$ | 0.98 | 0.88 | 0.88 | 0.00 | 0.02 | 0.03 |
| $p_{X,A=b,Y=0}(x)$ | 0.97 | 0.97 | 0.97 | 0.05 | 0.01 | 0.02 |
| $p_{X,A=b,Y=1}(x)$ | 0.97 | 0.91 | 0.95 | 0.03 | 0.06 | 0.05 |

The Adult, Law, and Dutch Census datasets respectively contain 48842, 20798, and 60420 samples. Using CT-GAN (Xu et al. (2019)), we train a generator to learn the underlying distribution of each dataset and use it to produce one million samples for each. Lloyd's algorithm (Linde et al. (1980)) can then be applied to these samples to construct a discrete approximation of the population distribution. To verify the fidelity of these samples, we use the Pearson Correlation Coefficient (PCC) and Total Variation (TV) Distance to compare the following discrete distributions constructed from the true and generated samples from each dataset: $p_{X,A=a,Y=0}(x), p_{X,A=a,Y=1}(x), p_{X,A=b,Y=0}(x), p_{X,A=b,Y=1}(x)$. We use the bound derived in Appendix A to determine how densely we can quantize our original data, while maintaining the fidelity of the distribution, $(X, A, Y)$, over each cell. Empirically setting the confidence and error parameters to $\delta = 0.95$ and $\Delta = 0.05$, this bounds suggests that we may only partition the true Adult, Law, and Dutch datasets into 48, 20, and 59 VQ cells, respectively. Thus, we use these specifications to construct discrete distributions from the true and generated samples for each dataset. Table 4 provides the resulting PCC and TV Distance results between the distributions created from the real and generated data. There

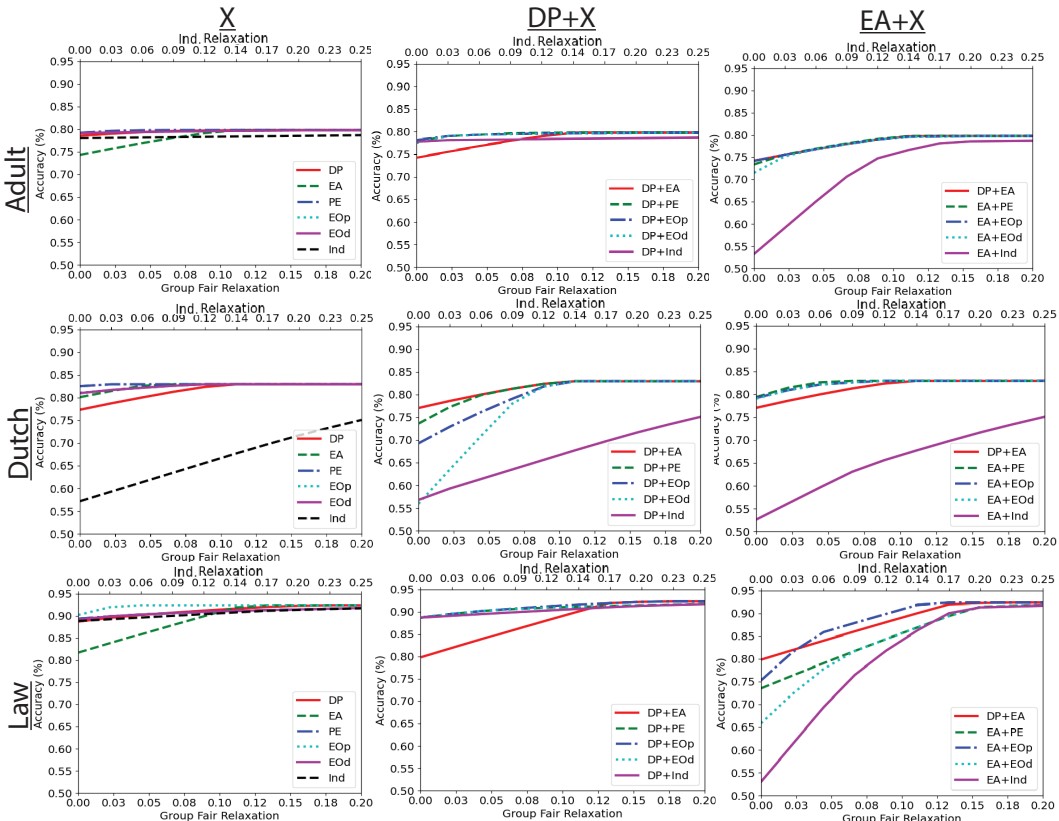

Figure 5: Pareto frontiers capturing the accuracy-fairness tradeoff for three datesets under unawareness of the sensitive attribute. Each plot provides curves for different pairings of fairness constraints; namely, DP, EA, PE, EOd, and Ind.

is good correlation between each distribution constructed from real and generated data, with all values having at least a PCC of 0.88, the majority of which are well above 0.90. Similarly, all TV Distance values are quite low with all values falling below 0.06, though most are less than 0.05. This suggests that generator has learned the population distribution associated with each dataset, providing us with confidence in using it to construct a more fine-grain cell decomposition from the large number of samples produced by the generator. We set $N_c = 256$ for all experiments conducted in the body of the main paper.

## K    FULL SET OF PARETO FRONTIERS FOR SENSITIVE-UNAWARE

Since spacing limitations prevented us from including more plots in Fig. 2 of the main paper, we provide the remaining Pareto frontiers under unawareness of the sensitive attribute that could not be fit into that figure in Fig. 5 below. As was the case for the sensitive-attribute-aware plots in Fig. 2, for the Adult and Law datasets, we see much smaller accuracy dropoffs in pairings involving DP as opposed to pairings involving EA. The converse is true for the Dutch Census dataset, further highlighting the distributional dependence of the tensions that exist among these fairness notions.

All Pareto frontiers discussed up to this point deal with no more than two combinations of fairness notions. In Fig. 6 we provide plots containing three or more combinations of fairness definitions. It can clearly be seen that the accuracy dropoffs in these plots are much more drastic than those from Fig. 2 or 5, as is to be expected. Noteably, enforcing local individual fairness on top of any combination of three group fairness prohibits any meaningful classification result with all accuracies dropping to around 50% with the inclusion of the Ind constraint inside this figure.

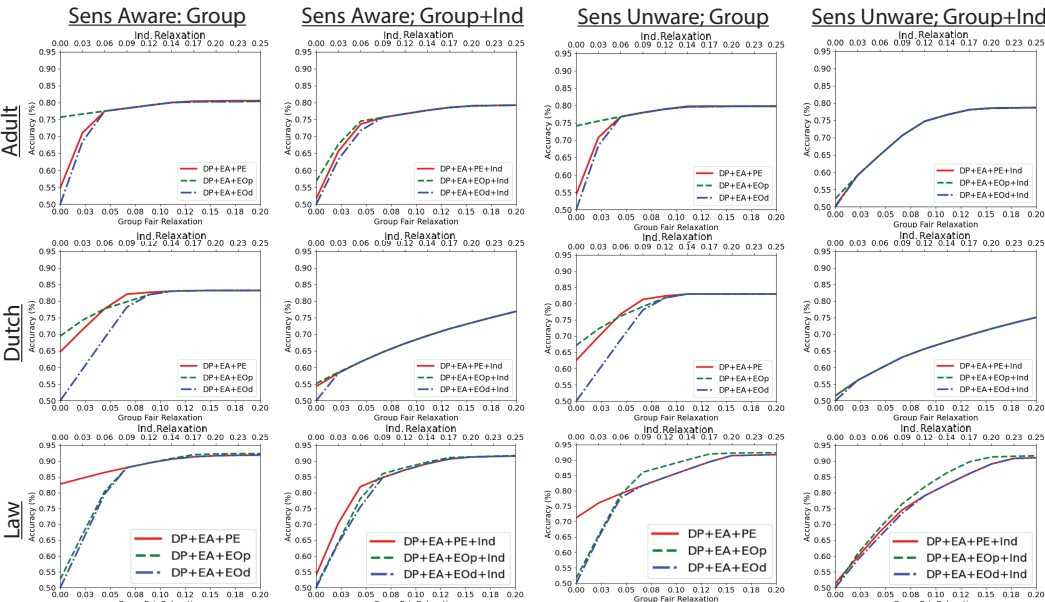

Figure 6: Pareto frontiers capturing the accuracy-fairness tradeoff for three datesets under awareness (columns 1 and 2) and unawareness (columns 3 and 4) of the sensitive attribute. Plots include pairs of three group fairness definitions with (columns 2 and 4) and without (columns 1 and 3) individual fairness constraints.

# L   VQ Granularity vs Time Complexity

Given that the generator has produced a sufficient number of samples for modeling the population distribution, fine-grain partitioning the space of feature vectors is desirable for precise analysis since increasing the number of VQ cells reduces their size, leading the average information associated with a cell centroid to be more representative of all the feature vectors within a cell. However, there is a tradeoff between the number of VQ cells and optimization time complexity. We selected 256 as the number of VQ cells for our analysis to strike a balance between precision and time complexity. In Table 5, we report the average accuracy and standard deviation of the Pareto frontiers for different combinations of group fairness constraints for relaxations running from 0 to 0.2. The results show that significantly increasing the number of VQ cells from 256 to 512 cells only leads to marginal changes in the accuracy and standard deviation of the Pareto frontiers for each dataset, indicating that the trends have begun to plateau. These results remain consistent across datasets. Hence, 256 cells provides a faithful representation of the fairness-accuracy tradeoff.

In Table 6, we report the time complexities associated with the different modules of our analysis. All experiments were run on a Macbook Pro (1.7 GHz Quad-Core Intel Core i7) with no GPU support. Training the codebook for the 256 VQ cell decomposition is the most expensive of all the tasks that we preformed, but the time complexities are reasonable, especially since training only needs to be performed once for each dataset. The results for the four remaining tasks in the table consist of averages and standard deviations of the computational complexities for solving each minimization problem in our optimization framework for all relaxations of each group and individual fairness combination reported in Figures 2 and 5 and Tables 2-4. The linear programming minimization problems are able to be solved extremely efficiently with runtimes all far below one second for processing. Optimization problems (10) and (35) are much more expensive since a large number of gradient steps must be applied for the solution to converge. The runtimes for these problems tend to be much smaller when only pairs of group fairness constraints are active since each group fairness definition is associated with just one constraint (in constrast to individual fairness). Still, the average runtimes for solving problems (10) and (35) are still typically

around 30 minutes, with worst case runtimes typically no worse than one hour and 30 minutes. Using more adaptive learning rate schedules could further improve these runtimes.

Table 5: Average accuracy of group fairness Pareto frontiers with relaxations varied between 0 and 0.2. Standard deviations of the accuracies of each frontier are provided in parentheses.

| | Adult | | | | | |
|---|---|---|---|---|---|---|
| | Awareness | | | Unawareness | | |
| # Cells | DP+EA | DP+EOd | EA+EOd | DP+EA | DP+EOd | EA+EOd |
| 16 | 0.734 (0.039) | 0.775 (0.000) | 0.720 (0.059) | 0.701 (0.062) | 0.775 (0.000) | 0.699 (0.065) |
| 32 | 0.763 (0.022) | 0.784 (0.004) | 0.750 (0.047) | 0.754 (0.029) | 0.781 (0.002) | 0.744 (0.047) |
| 64 | 0.776 (0.019) | 0.786 (0.005) | 0.768 (0.028) | 0.763 (0.026) | 0.782 (0.002) | 0.760 (0.032) |
| 128 | 0.778 (0.019) | 0.789 (0.005) | 0.772 (0.026) | 0.769 (0.025) | 0.786 (0.004) | 0.766 (0.030) |
| 256 | 0.789 (0.017) | 0.798 (0.008) | 0.785 (0.023) | 0.782 (0.020) | 0.793 (0.007) | 0.779 (0.026) |
| 512 | 0.803 (0.015) | 0.808 (0.012) | 0.798 (0.023) | 0.798 (0.018) | 0.805 (0.011) | 0.794 (0.026) |
| | Dutch Census | | | | | |
| | Awareness | | | Unawareness | | |
| # Cells | DP+EA | DP+EOd | EA+EOd | DP+EA | DP+EOd | EA+EOd |
| 16 | 0.688 (0.008) | 0.672 (0.042) | 0.697 (0.007) | 0.674 (0.001) | 0.657 (0.035) | 0.667 (0.012) |
| 32 | 0.728 (0.011) | 0.704 (0.057) | 0.735 (0.011) | 0.719 (0.007) | 0.695 (0.052) | 0.717 (0.010) |
| 64 | 0.791 (0.014) | 0.747 (0.082) | 0.797 (0.012) | 0.783 (0.014) | 0.741 (0.078) | 0.784(0.011) |
| 128 | 0.796 (0.015) | 0.752 (0.084) | 0.801 (0.015) | 0.790 (0.014) | 0.746 (0.081) | 0.787 (0.016) |
| 256 | 0.821 (0.013) | 0.766 (0.093) | 0.826 (0.010) | 0.814 (0.020) | 0.765 (0.093) | 0.822 (0.012) |
| 512 | 0.839 (0.014) | 0.778 (0.101) | 0.848 (0.007) | 0.829 (0.024) | 0.777 (0.100) | 0.846 (0.008) |
| | Law | | | | | |
| | Awareness | | | Unawareness | | |
| # Cells | DP+EA | DP+EOd | EA+EOd | DP+EA | DP+EOd | EA+EOd |
| 16 | 0.823 (0.049) | 0.891 (0.002) | 0.821 (0.052) | 0.803 (0.063) | 0.890 (0.002) | 0.765 (0.100) |
| 32 | 0.841 (0.046) | 0.894 (0.004) | 0.839 (0.048) | 0.821 (0.060) | 0.891 (0.003) | 0.791 (0.085) |
| 64 | 0.844 (0.045) | 0.896 (0.004) | 0.845 (0.046) | 0.823 (0.059) | 0.894(0.004) | 0.796 (0.070) |
| 128 | 0.849 (0.045) | 0.899 (0.006) | 0.853 (0.045) | 0.832 (0.057) | 0.897 (0.005) | 0.819 (0.072) |
| 256 | 0.891 (0.032) | 0.910 (0.010) | 0.876 (0.055) | 0.879 (0.045) | 0.907 (0.009) | 0.833 (0.085) |
| 512 | 0.902 (0.031) | 0.916 (0.013) | 0.892 (0.050) | 0.888 (0.044) | 0.914 (0.012) | 0.855 (0.077) |

Table 6: Average time complexity of the different optimization tasks in the proposed framework for each dataset.

| Task | Adult | | Dutch Census | | Law | |
|---|---|---|---|---|---|---|
| 256 VQ Cell Training | 5375s | (——) | 8513s | (——) | 8417s | (——) |
| Solving (6) | 0.007s | (0.007) | 0.007s | (0.007) | 0.007s | (0.008) |
| Solving (10) | 839s | (525) | 1348s | (941) | 1644s | (1823) |
| Solving (33) | 0.033s | (0.009) | 0.043s | (0.008) | 0.035s | (0.009) |
| Solving (35) | 2773s | (2327) | 2104s | (1855) | 1785s | (1743) |

## M    Summary of Notation

Table 7: Notation

| | | |
|---|---|---|
| bold face capital letter (e.g. $\mathbf{X}$) | $\triangleq$ | matrix |
| bold face lowercase letter (e.g. $\mathbf{x}$) | $\triangleq$ | column vector |
| $\mathbf{X}[i,j]$ | $\triangleq$ | element in $i^{th}$ row and $j^{th}$ column of $\mathbf{X}$ |
| $\mathbf{x}[i]$ | $\triangleq$ | $i^{th}$ element of $\mathbf{x}$ |
| $\mathbf{T}$ | $\triangleq$ | Matrix representing decorrelation mapping for feature vectors |
| $\mathbf{T}_a$ | $\triangleq$ | Matrix representing decorrelation mapping for feature vectors specific to Group $a$ |
| $\mathbf{T}_b$ | $\triangleq$ | Matrix representing decorrelation mapping for feature vectors specific to Group $b$ |
| abbreviation $ts$ | $\triangleq$ | Affiliation to test set |
| $A$ | $\triangleq$ | Sensitive attribute random variable |
| $Y$ | $\triangleq$ | Class label random variable |
| $X$ | $\triangleq$ | Feature vector random variable |
| $\mathcal{X}, \mathcal{A}, \mathcal{Y}$ | $\triangleq$ | Sample spaces for the feature vector, sensitive attribute, and class label random vector/variables |
| $S$ | $\triangleq$ | Randomized Scoring function |
| $\hat{Y}$ | $\triangleq$ | Class label estimator |
| $\mathbf{s}^B$ | $\triangleq$ | Scores produced by unconstrained Bayesian oracle |
| $\mathbf{s}^F$ | $\triangleq$ | Scores produced by fair Bayesian oracle |
| $\mathbf{m}$ | $\triangleq$ | deviation between scores produced by unconstrained and fair Baysian oracles (i.e. $\mathbf{s}^B$ and $\mathbf{s}^F$). We optimize for this vector. |
| $\mathbf{1}_k$ | $\triangleq$ | column vectors of length $k$ containing all 1s |
| $\mathbf{0}_k$ | $\triangleq$ | column vectors of length $k$ containing all 0s |
| $\mathbf{I}_M$ | $\triangleq$ | $M \times M$ identity matrix |
| $\mathbf{O}_{M,N}$ | $\triangleq$ | $M$ and $N$ matrix of all zeros |
| $\mathbf{1}_{M,N}$ | $\triangleq$ | $M$ and $N$ matrix of all ones |
| $\mathbf{p}$ | $\triangleq$ | Vector capturing distribution involving $X$ in which $X$ *is not* a variable on which we condition. E.g. The $i^{th}$ element of $\mathbf{p}_0^a$ is equal to $P(X = \mathbf{x}_i^c, A = a\|Y = 0)$ |
| $\mathbf{q}$ | $\triangleq$ | Vector capturing distribution involving $X$ in which $X$ *is* a variable on which we condition. E.g. The $i^{th}$ element of $\mathbf{q}_0^a$ is equal to $P(A = a\|X = \mathbf{x}_i^c, Y = 0)$ |