# OpenReview forum: "Balancing Fairness and Accuracy in Data-Restricted Binary Classification"
_ICLR.cc/2024/Conference — Submitted to ICLR 2024_

### Official Review · Reviewer_ENEz · 2023-10-27

**Soundness:** 3 good
**Presentation:** 3 good
**Contribution:** 2 fair
**Rating:** 5
**Confidence:** 3

**Summary:**

The authors conduct a theoretical examination of the conflicts that can arise between various fairness notions, encompassing both statistical and individual notions of fairness. This analysis is conducted across four distinct data availability scenarios, contingent on factors such as whether the sensitive attribute is known and whether the classification features need to be uncorrelated with it.
To facilitate this analysis, the authors initially create a discrete approximation of the joint distribution of features and labels by employing a generator that is later densely sampled. Subsequently, they address the optimization problem associated with the Bayesian oracle, taking into account a series of fairness constraints that are relevant to the desired fairness notions. This optimization problem yields the cost in terms of utility resulting from the chosen fairness constraints. The empirical findings provide insights into how different choices of these fairness notions impact utility.

**Strengths:**

S1 - The research question posed by the authors holds significant importance due to its potential implications and relevance in the field. Diverse fairness criteria are recognized to sometimes conflict with each other, and the appropriateness of a specific criterion relies on the specific context of the classification task. Consequently, the availability of a method for assessing the impact on utility when simultaneously considering multiple fairness criteria is of paramount significance.

S2 -The paper impressively maintains a high level of clarity in its writing, ensuring that complex concepts are presented in an accessible manner. The tone is professional and appropriate for the subject matter, which enhances the overall reading experience. The division of content into subsections aids in navigating the paper and finding specific information efficiently. Additionally, the notation used is consistently well-defined, making it easier for readers to grasp the mathematical aspects of the paper.


S3 - The paper not only discusses the computational aspects but also delves into a detailed examination of how individual components affect the overall computational requirements. This level of granularity in understanding the computational burden is crucial for practitioners and researchers looking to implement the proposed approach.

S4 - The paper's dedication to reproducibility is evident through its comprehensive and thorough documentation of the experimental setup. This includes precise descriptions of the experimental conditions, datasets, parameters, and methodologies used in the research.

S5 - The framework they introduce effectively addresses the research questions outlined in the introduction. Moreover, the paper consistently substantiates all its claims.

S6 - Figure 1 serves as a valuable and informative visual representation of the method. It significantly aids in understanding various aspects, particularly the dense sampling, VQ, and the delineation of the four scenarios. Nonetheless, there is room for improvement in terms of the image quality.

**Weaknesses:**

W1 - The statement 'Law and Adult datasets exhibit a tension in pairings with EA, while pairings with DP are more easily satisfied' calls for a deeper exploration into the underlying reasons. It's essential to investigate the dataset characteristics, including the distribution of class labels and other relevant factors, to understand why this tension arises. Moreover, it's important to analyze the mathematical implications of each fairness notion in the context of these datasets. What does it mean for a dataset to satisfy DP, and what about EA? Furthermore, exploring why certain fairness pairings are more compatible than others is crucial.

In the case of 'For example, strictly satisfying EA+PE causes an accuracy reduction of 3% between the awareness and unawareness situations for the Adult dataset. However, for the Law dataset, strictly satisfying EA+PE leads to an accuracy drop of 10% between the awareness and unawareness situations,' it's imperative to provide an in-depth explanation for these disparities. While you present numerical results stemming from the optimization problem, supplementing this with a detailed rationale is essential. This work can greatly benefit from utilizing the proposed mathematical framework to shed light on these questions. This ability to provide insights into why certain outcomes occur is a key strength of your research.

I firmly believe that the paper should not only present numerical results but should also delve into the reasons behind these results. Offering a comprehensive explanation supported by the mathematical framework is what can truly enhance the quality and depth of this work. This is particularly significant for a paper seeking publication in this conference, and your research indeed possesses the tools to provide such valuable information.


W2 - The authord have overlooked a significant paper that explores the Bayesian-optimal classifier under fairness constraints [1]. It would be beneficial if the authors could consider a comparison between their fair Bayesian oracle and the approach presented in this paper in the experimental section.

W3 - You've omitted the fairness notion that demands parity in positive predictive value, a well-known concept that often conflicts with the notion of equality of odds. It would be of great interest if you would also consider this definition inside your analysis.

W4 - I understand that space constraints can be a challenge, and I believe that the decision to place the cases where S is available in the Appendix, thus prioritizing a robust introduction to provide context and adequately introduce the problem formulation and setting, is a prudent choice. This approach is particularly valuable given that the primary paper addresses the more complex scenarios (when access to S is not available). However, it would be beneficial to provide a brief description of these shorter sections in the main text. In the event of acceptance, the authors could utilize the additional page provided to incorporate these sections into the main text. These could be presented as special cases, which are comparatively simpler, within the framework mentioned in the primary paper.

W5 - The authors opt for $\lambda = 15$ and $\beta = 25$ as parameter values (Section 4.2.), but they do not present alternative results based on different parameter selections. Furthermore, they do not provide an explanation for their choice of these values or an assessment of the impact that varying these values may have on the results. In other words, there is a lack of exploration regarding the sensitivity of the results to changes in these parameter values.

W6 - It remains unclear, and there is a lack of in-depth analysis regarding whether achieving decorrelation consistently results in minimal accuracy trade-offs, or if specific dataset configurations are required for this to hold true.

W7 - Page 4 typo → ‘subscripts (subscripts)’


[1] Zeng, X., Dobriban, E., & Cheng, G. (2022). Bayes-optimal classifiers under group fairness. arXiv preprint arXiv:2202.09724.

**Questions:**

Q1 -You suggest a potential generalization to non-binary sensitive attributes through combinatorial extension. It's important to explore to what extent this would impact the framework you've outlined. Can this extension be seamlessly applied, or are there numerous challenges to overcome?Additionally, when dealing with non-binary sensitive attributes, how would you approach the comparison of group notions? Would you consider using maximum disparity as a measure? If so, would this necessitate a modification of the constraints, potentially rendering the problem non-convex?

Q2 - To what extent are the results influenced by the quality of the discrete approximation produced by the generator G?

Q3 - You mentioned that for the formulations in Section 3.1 and 3.2, sensitive information is not required. However, how do you then assess the constraints, such as (3), which involve considerations like $\boldsymbol{p}_a$ and $\boldsymbol{p}_b$?

Q4 - You conclude that 'This suggests that the accuracy drop-off of a fair classifier is less dependent on how correlated the features are with the sensitive attribute and more dependent on the strictness of the fairness enforcement.' Does this observation apply solely to the Law dataset, or is it a general trend across all datasets? Is it consistently valid, or are there specific configurations where this observation might not hold true?

Q5 - Have you explored the application of three or more fairness notions? How does the method's performance scale as the number of constraints increases?

Q6 - Why was the setting of $\lambda = 15$ and $\beta = 25$ chosen (Section 4.2.)? Additionally, how responsive are the results in Section 4.2 to alterations in these parameter values?

---

### Official Review · Reviewer_5mdp · 2023-10-27

**Soundness:** 2 fair
**Presentation:** 3 good
**Contribution:** 2 fair
**Rating:** 5
**Confidence:** 4

**Summary:**

The paper introduces a novel approach, where a fair decision is made based on cluster memebership. Various metrics are considered and the method allows to study trade-off between accuracy and fairness measures. It also allows using several fairies measures at the same time.

The beginning of introduction that describes the general fair classification problem reads very well and can be even educational for a general reader. The paragraph that describes the method of the present paper is not well written. It is not clear why a general fairness problem motivates to do quantization. It is also not clear what is Bayess oracle. Is it relevant to Bayess-optimal classifier? I recommend to refer to paper [1] who study optimal classifier for fairness-accuracy trade-off, perhaps it will make motivation stronger. What is the meaning of expression latent structure here?

Overall, it looks to me that the method suggests a non-parametric estimation for p(Y = 0, 1 | X). Classification and feature generation based on quantisation is not novel, to the extent that it is a popular trick on Kaggle. However, it have not been studied in the fairness context, and can bring a fresh perspective to post-hoc methods that only take into account prediction and sensitive attribute [2]. My concern is that in the presented form it is not really a post-hoc method and it is applied directly to features X. It seems that it can harm the accuracy dramatically, and it indeed looks so from the experiments in Fig. 2. For instance, for Adults, the accuracy of unconstrained should be around 83%, when using a small MLP.

It is also not clear, how well your method approaches the actual Pareto-Frontier. For example, there is no comparison to baselines from existing literature. Penalization-based methods like [4] naturally offer some trade-off, but require retraining.

Lastly, I mention some confusion of terms. In p. 3 you write "Let S : X → [0, 1] be a randomized scoring function". A "score" function typically represents evaluation of p(Y = 1| X), and then it is thresholded. It is not used directly to output a randomized classifier, since it is would be suboptimal. I am not against randomised classfier, since it can help fairness, the problem comes from using the word "score" which can be misleading. Furthermore, the authors misuse the term Predictive Equality [3]. In the original formulation, the conditioning must happened on $\hat{Y} = 1$. Also, your Equal Accuracy is equivalent to Demographic Parity in the binary case.

[1] Menon & Williamson (2019) The cost of fairness in binary classification

[2] Hardt et al. (2016) Equality of opportunity

[3] Chouldechova (2017) Fair prediction with disparate impact: A study of bias in recidivism prediction instruments

[4] Zafar et al (2017) Fairness constraints: Mechanisms for fair classification.

**Strengths:**

The idea of using clusterization for fair classification is new.

**Weaknesses:**

Although the use of VQ is interesting, it is poorly motivated in the introduction. Also, it would help including some references that study the trade-off.

Misuse of terms like score and predictive equality.

I think the method could be stronger as a post-hoc method. Compare to Hardt et al (2016) who make a randomized decision based on (\hat{Y}, A), where $\hat{Y}$ is unconstrained classfier. Essentially, they treat all elements in each of four groups $(\hat{Y} = y, A= a)$ equally. Your method can allow to cheaply push it forward, making the decision depend on $X$ as well.

**Questions:**

Can the analysis be generalised somehow to one abstract case? Similar to Kim eta al. "FACT: A Diagnostic for Group Fairness Trade-offs".

---

### Official Review · Reviewer_VFAT · 2023-10-28

**Soundness:** 3 good
**Presentation:** 3 good
**Contribution:** 1 poor
**Rating:** 3
**Confidence:** 3

**Summary:**

This paper empirically examines and plots the tradeoff between accuracy and combinations of fairness criteria on several datasets, under the aware and unaware settings and added constraint that the representation is independent from the sensitive attribute.  The results are generated by treating the dataset as equal to the population, then solving the label assignment problem maximizing the accuracy subject to fairness.

**Strengths:**

- The paper is well-written and easy-to-follow.
- One of the setting considered—learning fair representations where the feature map cannot depend on the sensitive attribute (unaware)—is interesting.

**Weaknesses:**

1. This paper aims to investigate the tradeoff between accuracy and combinations of fairness criteria, and the study is carried out by plotting the achievable tradeoff on several datasets.  However, I fail to grasp the main message from the results.

	- What are the key observations?  Are the observations universal or data dependent?
	- Most importantly, in which scenarios would the cost of fairness (i.e., tradeoff) be small/large?  I do not think this question can be answered without a theoretical study, e.g., as done in Hardt et al. (2016) and Zhao and Geoffrey (2022).

2. I do not see the practical implications of the results, i.e., how would they "help developers of fair ML models", since the author did not answer the important question of how to achieve the tradeoff in practice.

	How to train a generalizing classifier/score function that attains the plotted tradeoff, not just on the dataset (training set), but on the population (during inference)?  Also, how to learn the feature mapping in the "decorrelated setting" without resorting to vector quantization, which introduces approximation error and estimation error that are potentially exponential in the dimension.

3. The use of a generator and vector quantization in the experiments seems ad-hoc and unprincipled.

	- Vector quantization is used to "partition" the input space so that the task of finding a fair representation can be represented as a optimization problem with a finite number of variables (i.e., we are getting rid of complications related to approximation).  But quantization introduces approximation error, which means results generated involving VQ do not really capture the fundamental tradeoff.  This limitation is also not mentioned.
	- I do not understand the purpose of the generator.  Does the generator converge to the true underlying distribution?  If not, what is the rate?  Also, using the distribution learned by the generator on finite data would introduce estimation error, which is also not discussed.
	- Instead of learning a generator, why not just use simple nonparametric estimators (e.g., KDE)?
	- The authors could have approached this problem from an information-theoretic perspective, see, e.g., "Fundamental Limits and Tradeoffs in Invariant Representation Learning" by Zhao et al. (2022).

Minor: $\mathbf{p}^y$ in eq. (1) is not defined, and the formatting does not conform to ICLR guidelines.

**Questions:**

See weaknesses.

---

### Official Review · Reviewer_wq6f · 2023-11-02

**Soundness:** 1 poor
**Presentation:** 2 fair
**Contribution:** 1 poor
**Rating:** 3
**Confidence:** 4

**Summary:**

The paper studies the problem of fair learning under restricted access to sensitive attributes. They consider four scenarios based on 1) whether sensitive attributes are present in the data, and 2) whether or not the predictions should be decorrelated with the sensitive attributes. For each setting, they formulate the fair learning problem as a convex constrained optimization problem and provide experimental results for the Pareto frontier of fairness vs. accuracy on Adult, Law, and Dutch Census data sets.

**Strengths:**

-The problem of fair learning with restricted access to sensitive attributes is important because it has practical application. This paper introduces different restriction models on sensitive data that could be helpful for fair machine learning research.

-The paper provides experimental results on a variety of data sets to examine the algorithms.

**Weaknesses:**

-The first step of the proposed method is forming a discrete approximation of the data distribution using a generator, which is then followed by partitioning the data space. This, in turn, allows the authors to assume that the probability mass on every cell of the partition can be approximated by the probability mass of the centroid of the cell (why?). While I do not quite follow these steps and am confused by it, I don’t even know why these steps are needed in the first place. Standard machine learning approach is to provide guarantees on a data set sampled form a distribution and then appeal to generalization guarantees to get results that hold over the distribution.

-The metric that is used to compare the accuracy of two models (on page 5) seems flawed: according to this metric, if $p^1$ and $p^0$ are equal (all $1/2$ vector), then every two models have the same accuracy, which doesn’t make any sense. The same goes with the way that the fairness notions are formulated. For example, for the case of demographic parity (DP), if $p_a$ and $p_b$ are equal, then the proposed notion suggests that DP is always satisfied. But DP depends on the conditional distribution of classifications, conditioned on group memberships, i.e., we can construct an example for which $p_a$ and $p_b$ are equal but DP is not satisfied. Overall, I am not quite sure if the problem formulation in this paper, in particular notions of accuracy and fairness, are mathematically correct.

-The technical parts of the paper are hard to read mostly because the notations are difficult to parse.

-The discussion of related works is not complete. In particular, Jagielski et al. (2019) (“Differentally Private Fair Learning”) focus on the similar problem of achieving fairness when there is limited access to sensitive attributes. This is a very relevant paper and should be discussed in the related works.

**Questions:**

See weaknesses.

---

### Meta-Review · Area_Chair_ikhx · 2023-12-01

**Metareview:**

The paper follows a list of known results on the tension between different notions of fairness. The current study is formalized in an optimization problem that allows to derive the utility cost of competing notions of fairness when an idealized Bayesian oracle is applied. The analysis is carried out experimentally on several datasets.

The paper does not make a clear contribution to the state of the art. It needs further analysis of the experimental observations and certainly needs to be grounded in a theoretical analysis.

The authors did not provide a response to the reviewers.

**Justification For Why Not Higher Score:**

The authors did not provide a response to the reviewers.

**Justification For Why Not Lower Score:**

N/A

---

### Decision · Program_Chairs · 2024-01-16

Reject